ecology/climatology

tree-ring width, stable carbon isotope, Masson pine, subtropical China

**Author for correspondence:**
Jian Wang
e-mail: wangjian@njnu.edu.cn

# Comparison of dendroclimatic relationships using multiple tree-ring indicators (tree-ring width and $\delta^{13}$C) from Masson pine

Hongliang Gu[1,2], Jian Wang[1], Chao Lei[1] and Lijuan Ma[1]

[1]School of Geography, Nanjing Normal University, Nanjing 210023, People's Republic of China
[2]School of resources and environment, Anqing Normal University, Anqing 246011, People's Republic of China

  HG, 0000-0001-6441-5991; JW, 0000-0002-9102-6518

This study addressed the effects of climate drivers on the tree-ring width (TRW) parameters (total ring width (TR), earlywood width (EW) and latewood width (LW)) and the total ring $\delta^{13}$C series of different wood components (whole wood, α-cellulose and holocelluose) from Masson pine in subtropical China. Pairwise correlation coefficients between three ring width parameters were statistically significant. EW and LW did not reveal much stronger climate sensitivity rather than TR. This indicated that the use of intra-annual ring width has little benefit in extracting more climate information. The mean $\delta^{13}$C series of the three components of the total ring had the strongest climate response to the July–September relative humidity ($r = -0.792$ (whole wood), $-0.758$ (holocellulose) and $-0.769$ (α-cellulose)). There are no significant differences in the dendroclimatic relationships of the $\delta^{13}$C series of different wood components. Through both stationary temporal and spatial-statistical perspectives, the moisture drivers (summer/autumn) had a significant impact on three ring width parameters and three components of Masson pine. Overall, the radial growth and the $\delta^{13}$C series showed different responses to the same climate drivers during the same period. Moreover, the R-squared values of the strongest climate-proxy correlation coefficients were smaller than 50% for TRW. Consequently, the $\delta^{13}$C series of Masson pine may be a more representative climate proxy than TRW parameters for dendroclimatology in subtropical China.

# 1. Introduction

Subtropical China (22°–34° N, 98°–122° E) is an important socio-economic development area, with high population density and complex biodiversity [1]. In recent years, due to climate change, extreme climate events, such as floods and freezing disasters, have occurred frequently, posing risks to public life and property in this region [2,3]. Therefore, assessing the long-term impact of climate and environmental change on forest dynamics in this region is beneficial to our work on climate disaster risk management. According to the advantages of long-term records, high resolution and wide spread, tree rings have been proven to be a useful natural archive for climate change research [4–7]. At present, dendroclimatology research works are increasingly carried out using tree-ring width (TRW) in subtropical China. One of the most frequently studied species is Masson pine [2,8–15]. It is one of the most widely distributed and abundant tree species in subtropical China [16,17]. However, due to the warm and humid environment, subtropical forests and woodlands have highly diverse developing plants and diverse wood anatomy. So the relationship between tree rings and climate drivers is complex [18–20]. Hence, to some extent, dendroclimatology research in subtropical China has been hampered thus far.

Progress in dendrochronology techniques allows for using additional tree-ring indicators to enhance the climate signal and extend the reconstruction season or reconstruct different climate environmental variables [21]. For example, compared with the total ring width (TR), earlywood width (EW) and latewood width (LW) have higher temporal resolution than TR (EW + LW) and may show more valuable climate and environment signals [22,23]. Some studies have shown that cross-dating was statistically more significant with LW than with TR [24], and LW appeared to have the strongest sensitivity to climate drivers [22,23,25]. Other researchers also reported that the EW revealed the strongest correlation with the early summer hydroclimatic signal [26,27].

On the other hand, compared with TRW, the stable carbon isotope ratio ($\delta^{13}$C) of tree rings has become a more reliable and representative climate proxy for palaeoclimate, palaeoenvironment, and global change ecology research in humid and warm areas [28–30]. However, due to the complex mixture of different organic compounds (whole wood (WW), holocellulose (HC) and α-cellulose (AC)) in tree rings, there are still differing opinions on which component is best for the extraction of climate change data [31–34]. Studies have shown that $\delta^{13}$C series of WW are better at extracting climate information than other organic compounds [32]. In comparison with other components, some researchers believe that cellulose has a unique composition, is easy to extract [35,36] and is more sensitive to climate change [37,38].

Consequently, to trace past climate change, conducting multiple tree-ring parameters should be widely developed in subtropical China. To our best knowledge, there has been little previous research performing a comprehensive analysis of TRW (TR, EW and LW) and targeting the $\delta^{13}$C series of different components of Masson pine. Additionally, we found that the boundaries of EW and LW from Masson pine are easily distinguishable. Thus, it is feasible to investigate the potential of the multiple proxies (TR, EW, LW, WW, HC and AC) of Masson pine for dendroclimatology studies. The questions we addressed are as follows: (i) How about the statistical and climate sensitivity properties of different TRW parameters from Masson pine? (ii) Are there differences in the dendroclimatic relationships of the $\delta^{13}$C series of different wood components? (iii) Between the $\delta^{13}$C of the total ring and the radial growth merits, which indicator is more sensitive to climate?

# 2. Material and methods

## 2.1. Study area

The mapped distribution of Masson pine forests in subtropical China used in this study is presented in figure 1a. Masson pine spatial distribution raster datasets were extracted from the vegetation map of China (scale 1 : 1 million). In the northern Masson pine spatial distribution region, the Tongbai Mountains (31°49′–32°24′ N, 113°16′–114°03′ E, the highest elevation is approximately 1794 m.a.s.l.) are located at the junction of Henan and Hubei provinces near the northern margin of the Masson pine distribution. The northwest border is the Nanyang Basin, and the southeast border is the Dabie Mountains, which are more than 120 km long. The interior of the Tongbai Mountains experience a continental monsoon climate zone, have brown and grey soil, and are dominated by native Masson pine forest. The woodland area is 839 km², and the forest coverage rate attains 38.7%. The annual mean temperature is 15°C. January is the coldest month (2°C, figure 1b), and July is the warmest

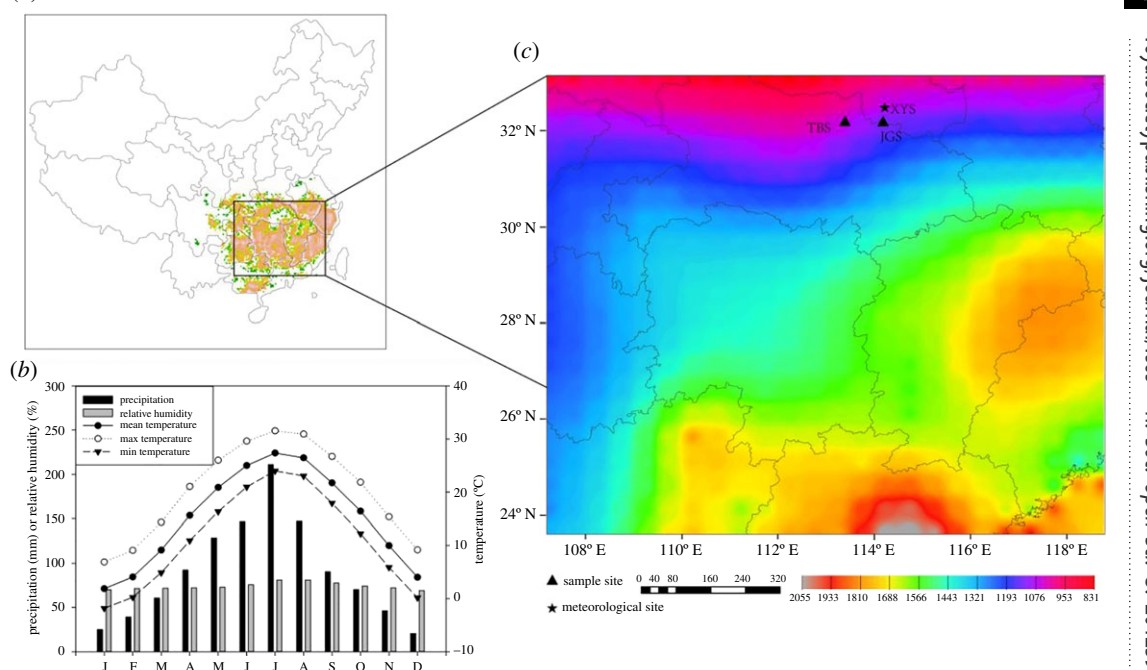

**Figure 1.** Study area characteristics. (*a*) Masson pine distribution [17]. (*b*) Mean monthly climate drivers (temperature, minimum temperature, maximum temperature, precipitation and relative humidity) in the study area. (*c*) Distribution of the long-term annual precipitation total over subtropical China. The black star shows the meteorological station, and black triangles show the sampling area ('TBS' denotes Tongbaishan, 'JGS' denotes Jigongshan).

month (28°C). The long-term annual mean precipitation is 1168 mm in this region. The mild temperature and long daylight in summer are beneficial for the growth of plants. Because the north region of the Tongbai Mountains lacks alpine barriers, cold-air outbreaks from the north can move straight through this region in winter and the cold winter is relatively long.

## 2.2. Field sampling and tree-ring width chronology construction

At each sampling area (Tongbaishan (32°24′ N, 113°16′ E) and Jigongshan (31°49′ N, 114°03′ E)), we selected two sites from shady and sunny slopes for tree-ring sampling. In total, 65 trees (mean diameter at breast height was approx. 1.3 m) were selected from edge trees or isolated trees with low canopy density in forests to reduce the impact of low-frequency changes in the TRW series due to tree competition. At approximately 1.2 m above the ground on both sides of the tree stem parallel to the direction of the slope, a 5.5 mm diameter increment corer was used to sample trees without conditions such as heart rot and scars. After drying them in the ambient air, the tree-ring cores were polished to facilitate ring identification [39]. A skeleton plot was first used to validate the preliminary calendar age of each tree ring, omitting the parts of the disc showing pseudo or absent tree rings [40]. Afterwards, TRW was measured using the AcuRite measuring system (0.001 mm) (VoorTech, NY, USA). Finally, the COFECHA program was used to identify the quality of the cross-dating [41,42], and TRW chronologies were calculated by the ARSTAN program [43]. The trend in tree growth was detrended by using the traditional method (i.e. negative exponential curve) [40].

## 2.3. Stable carbon isotope pretreatment and determination

To split the rings to obtain the stable carbon isotope signatures, tree-ring cores (12 mm diameter increment) were obtained from four relatively old trees in the study area, that is, trees with relatively longer TRW chronologies [31]. Sequentially, the tree cores were placed on clean glass plates for rasping. Four rasped increment cores of the same ring per tree were cut and mixed together. Then, they were packaged in clean Kraft paper bags and numbered accordingly. Each tree-ring sample was obtained by carving out individual whole rings, drying them for 3 days at 70–80°C and then grinding them to pass through a

100-mesh sieve. Part of each sample was collected for the determination of the $\delta^{13}C$ signal of WW, and the rest was used for cellulose extraction. HC and AC from wood samples were extracted following the standard procedures [44–47]. Finally, the isotope $^{13}C/^{12}C$ ratios were determined with a Thermo Finnigan-Deltaplus XP isotope mass spectrometer (IRMS). The $^{13}C/^{12}C$ ratios, referenced against the standard $\delta^{13}C$ (PDB), were expressed as $^{13}Cp$ in ‰ [48]. The analytical precision was typically within ± 0.1‰. The cellulose in individual tree-ring samples was analysed separately. Compared with the pooling method (mixing wood of the same year from several trees) [49], the calculation of the average value of $\delta^{13}C$ after measuring the $\delta^{13}C$ values from individuals may offer 'representative' isotopic values [50]. Therefore, the $\delta^{13}C$ values for each individual were measured, and the synthetic sequence of $\delta^{13}C$ in each study area was calculated using the arithmetic average method. Because the atmospheric $\delta^{13}C$ concentration has exhibited a significant downward trend since the beginning of the Industrial Revolution, we also mathematically removed the impact of atmospheric $\delta^{13}C$ on tree-ring $\delta^{13}C$ [51,52].

## 2.4. Climate material

To obtain a complete and accurate climate dataset, the Xinyang meteorological station (32°08′ N, 114° 03′ E, 114.5 m.a.s.l.) was used in the dendroclimatic analysis. The station is located near the sampling sites and has complete meteorological data (http://data.cma.cn/). The distance between the Xinyang station and the sample sites is approximately 53 km (figure 1c). The station is located at the point in this region where the difference between the accumulated annual and monthly temperature is the largest, the annual average daylight duration is the longest and the relative humidity is the lowest. The peak of relative humidity appears in August, the annual average precipitation is 1097 mm and the frost-free period is 227 days. December is the driest month (21 mm), and May–August is the wettest period, especially when the monthly sum precipitation reaches the maximum (211 mm) in July.

## 2.5. Statistical analysis methods

The statistical analyses of the tree-ring chronologies were performed in the dplR package [53]. To determine the climate drivers affecting the annual radial growth of Masson pine, simple and partial correlations [54,55] were carried out on tree-ring proxies and climate records from the previous June to the following September. Correlation analysis was performed with the first-order difference of the time series to check whether the significant correlation was caused by an actual trend or by the high-frequency variations of the time series. The significance of each correlation was evaluated using the bootstrapping method (1000 Monte Carlo simulations) [54]. Based on the correlation analysis, the main climate factors affecting the radial growth of trees in the study area were determined.

Additionally, a seasonal correlation analysis was carried out, and its temporal stability was tested by running a 35-year moving window with a 4-year offset [55]. To avoid a shortage of samples and the 'juvenile effect', the calendar period AD 1982–2011 was selected for $\delta^{13}C$ determination in the Tongbai Mountains. Finally, spatial correlation analysis was carried out with CRU TS4.02 grid data (http://climexp.knmi.nl) to explore the spatial significance of tree-ring proxies for climate drivers in this study area.

# 3. Results

## 3.1. Statistical characteristics of three ring width parameters

The statistical characteristics of the chronologies of Masson pine tree rings are shown in table 1. The time span of the tree-ring samples ranged from AD 1911 to 2013. The mean width (1294.02 µm) of EW was larger than that of LW (938.55 µm) in Masson pine. Overall, EW had the most variable signal as indicated by the standard deviations, compared with the corresponding means (table 1). The average sensitivity of TRW was approximately 0.2. In addition, the average sensitivity of the EW and LW chronologies was only slightly higher than that of the TR chronology. The signal-to-noise ratio, expressed population signal, and variance of PC1 were generally high for each tree-ring parameter, whereas LW had the lowest statistical values (table 1). Overall, the quality of the tree-ring chronologies in our study area met the requirements for the tree-ring climatology research.

Generally, running-window statistics were used to assess the temporal patterns of the signal strength from three tree-ring parameters. The values of Rbar were below the 0.2 level for TR and LW during AD 1960–1970, while they exceeded this level for EW over the entire period (figure 2). Additionally, EPS

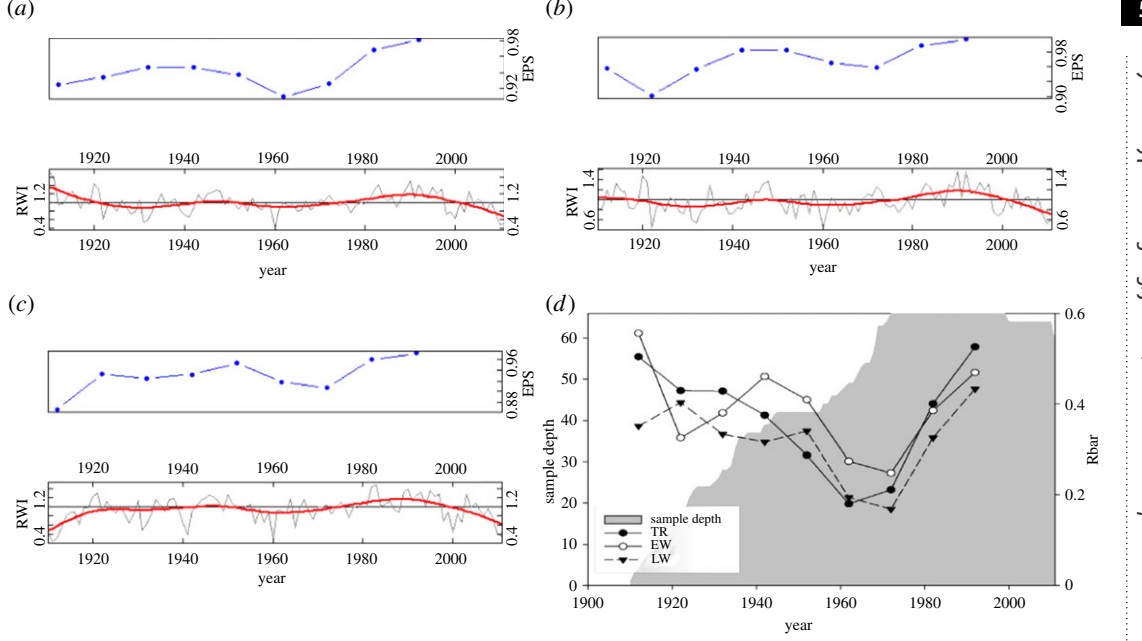

**Figure 2.** The chronologies and signal strength statistics of Masson pine. (*a*) TR, (*b*) EW, (*c*) latewood width (LW) and (*d*) mean inter-series correlation (Rbar) and sample depth.

**Table 1.** Statistical characteristics of TRW chronologies. Note: MW (mean width), MS (mean sensitivity) [56], SD (standard deviation), SNR (signal-to-noise ratio) [57], AC1 (intercorrelation), AR1 (first-order autocorrelation), EPS (expressed population signal) [58], PC1 (variance of PC1) and Gini (Gini difference) [59].

|  | MW (μm) | MS | SD | AC1 | AR1 | SNR | EPS | PC1 (%) | Gini | absent rings |
|---|---|---|---|---|---|---|---|---|---|---|
| Tongbai Mountains | | | | | | | | | | |
| EW | 1294.02 | 0.266 | 0.631 | 0.563 | 0.773 | 30.806 | 0.969 | 37.7 | 0.254 | |
| LW | 938.55 | 0.231 | 0.240 | 0.576 | 0.425 | 23.423 | 0.959 | 32.6 | 0.292 | 0.021% |
| TR | 2232.57 | 0.190 | 0.679 | 0.608 | 0.686 | 33.799 | 0.971 | 40.3 | 0.238 | |

**Table 2.** Pairwise computed correlation coefficients of TRW/$\delta^{13}$C chronologies.

|  | TR | LW | WW | HC |
|---|---|---|---|---|
| EW | 0.94 | 0.53 | — | — |
| LW | 0.88 | — | — | — |
| AC | — | — | 0.91 | 0.91 |
| HC | — | — | 0.94 | — |

values also validated the above results for the three tree-ring proxies. The EPS values for the three tree-ring parameters fluctuated over time but mostly ran together well above the critical threshold (0.85) [60]. Consequently, EW seems to remain the most stable and robust with time. In addition, EW makes up approximately 57.96% of the mean annual TR, which means that the TR signal may be dominated by EW variability. The correlation between TR and EW had the highest coefficient (0.94), and that between TR and LW revealed a weaker relationship (0.88) (table 2). The correlation coefficient for the relationship between EW and LW was 0.53.

## 3.2. The statistical characteristics of $\delta^{13}$C records

All $\delta^{13}$C sequences of WW were more depleted than those of HC and AC within individual trees according to two statistical indices (mean, standard deviation) (table 3), because the lighter lignin in

**Table 3.** The statistic analysis of $\delta^{13}C$ records of different components within individual trees ('SP01' denotes the sampling sites).

| | SP01 | | | SP02 | | | SP03 | | | SP04 | | |
| --- | --- | --- | --- | --- | --- | --- | --- | --- | --- | --- | --- | --- |
| | WW | HC | AC | WW | HC | AC | WW | HC | AC | WW | HC | AC |
| mean (‰) | −26.407 | −24.977 | −25.078 | −25.699 | −24.205 | −24.297 | −26.761 | −25.560 | −25.624 | −25.970 | −24.667 | −24.615 |
| median (‰) | −26.426 | −24.976 | −25.010 | −25.736 | −24.301 | −24.270 | −26.864 | −25.545 | −25.786 | −25.968 | −24.555 | −24.588 |
| max (‰) | −25.522 | −23.692 | −23.726 | −24.593 | −22.872 | −23.144 | −25.507 | −24.160 | −24.185 | −25.083 | −23.168 | −23.299 |
| min (‰) | −26.920 | −25.793 | −26.294 | −26.266 | −24.948 | −25.057 | −27.490 | −26.500 | −26.750 | −26.786 | −25.839 | −25.781 |
| s.d. (‰) | 0.330 | 0.441 | 0.515 | 0.395 | 0.484 | 0.425 | 0.457 | 0.575 | 0.624 | 0.438 | 0.611 | 0.551 |

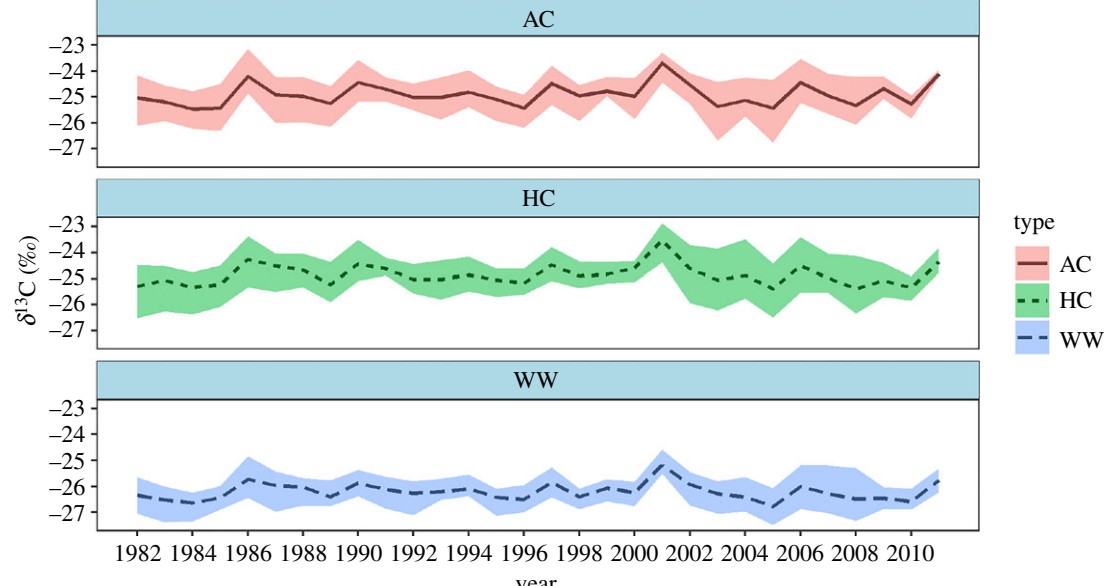

**Figure 3.** The established $\delta^{13}$C chronologies of different components from Masson pine (different line types denote the mean $\delta^{13}$C records for year by year).

wood was extracted from the HC samples but not from the WW samples. The values of WW ranged from −27.49 to −24.593‰, and AC and HC had ranges of −26.75 to −23.144‰ and −26.5 to −22.872‰, respectively. Although there were absolute differences in the values for WW, HC and AC, the sensitivity of different tree-ring components to climate change is evaluated based on the difference between the various trends in the carbon isotope components and the various trends in the climate drivers. Therefore, the differences in the trends of the $\delta^{13}$C series were used to judge whether each component changed based on the climate or not. The phase direction changes among the $\delta^{13}$C series of the three components of the Masson pine chronologies were highly consistent, i.e. they showed consistent upward or downward trends over time (figure 3). The computed pairwise correlation coefficients ranged from 0.91 to 0.94 (table 2). Compared with the incremental proxies, the $\delta^{13}$C series showed much stronger common variability. Moreover, the raw measured data did not display a prominent non-climate downward trend (figure 3), indicating that there was no need for a correction procedure. In addition, to measure the correlation between the previous values and the current value, an autocorrelation function was used to indicate the most useful past values for the current record (figure 4). The value at lag 1 indicates that each series value is correlated with the previous value, and so on. We can see that the lags do not have a significant effect within the confidence bounds, and only at lag 0 does the value appear significant. These findings show that the current value of the $\delta^{13}$C series cannot be obtained using previous values. Consequently, the previous climate drivers were not taken into account in subsequent climate response analyses.

## 3.3. The relation between tree-ring width and key climate drivers

### 3.3.1. Correlation between tree-ring width and monthly climate drivers

The significant correlation between the first-order difference of time series indicated that the tree ring of Masson pine reliably recorded climate signals (electronic supplementary material, figure S1). TR and EW were significantly negatively correlated with the previous mean temperature (i.e. that in June, August, September and October), and only the previous September mean temperature had a statistically significant negative relationship with the three tree-ring parameters (TR (−0.296, $p < 0.05$), EW (−0.220, $p < 0.05$) and LW (−0.349, $p < 0.05$)) (figure 5). Strong climate signals from the previous August maximum temperature (TR (−0.294, $p < 0.05$), EW (−0.287, $p < 0.05$), and LW (−0.215, $p < 0.05$)) were also observed in the three parameters. The previous August minimum temperature had a significant negative influence on TR (−0.192, $p < 0.05$) and EW (−0.218, $p < 0.05$), but TR and LW also showed significant relationships with the current March and July minimum temperatures. The three TRW parameters did not have a statistically significant positive relationship with single-month precipitation,

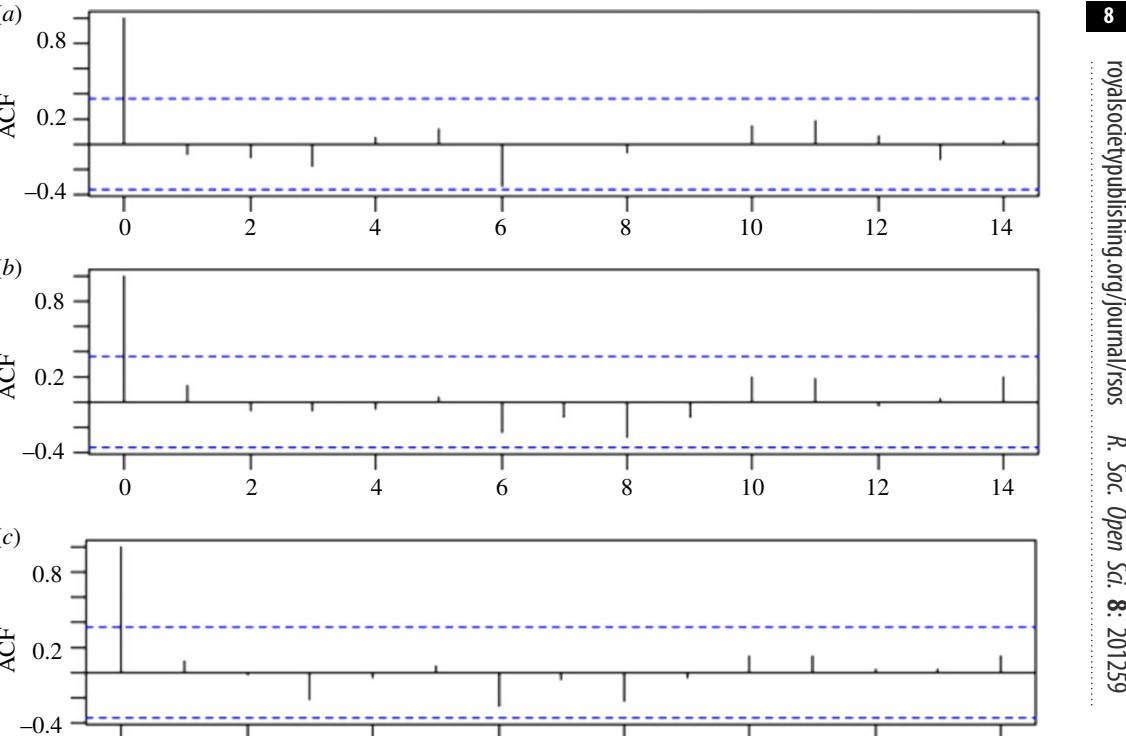

**Figure 4.** The autocorrelation function estimation of $\delta^{13}$C series in three components. Note: WW (*a*), AC (*b*) and HC (*c*).

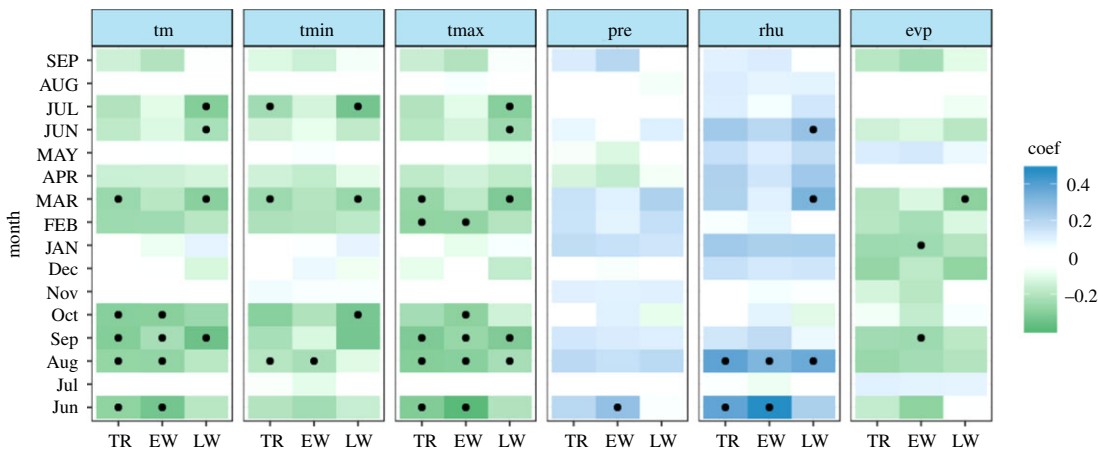

**Figure 5.** The bootstrapped correlation coefficients between TRW (TR, EW and LW) and monthly climate drivers from Xinyang meteorological station. Note: black dots indicate the significance level ($p < 0.05$). 'tm' denotes the mean temperature, 'tmin' denotes the minimum temperature, 'tmax' denotes the maximum temperature, 'pre' denotes the total precipitation, 'rhu' denotes the relative humidity and 'evp' denotes the evaporation. 'Jun' denotes the previous June and 'JUN' denotes the current June.

except LW. The previous June precipitation had a significant positive effect on LW (0.281, $p < 0.05$). However, overall, there was a positive relationship between the three width indices and monthly precipitation. The three width indices were positively correlated with relative humidity; TR (0.386, $p < 0.05$), EW (0.328, $p < 0.05$) and LW (0.360, $p < 0.05$) were significantly and positively correlated with the relative humidity in the previous August, and compared with TR (0.374, $p < 0.05$) and EW (0.481, $p < 0.05$), the relatively weak sensitivity of LW (0.173, $p < 0.1$) to the relative humidity in the previous June can be seen.

The correlation between the three width parameters and evaporation was weak, indicating that evaporation has a weak influence on the annual growth of Masson pine in this study area. As could be expected from the high correlation coefficients between EW and TR, fairly similar climate response

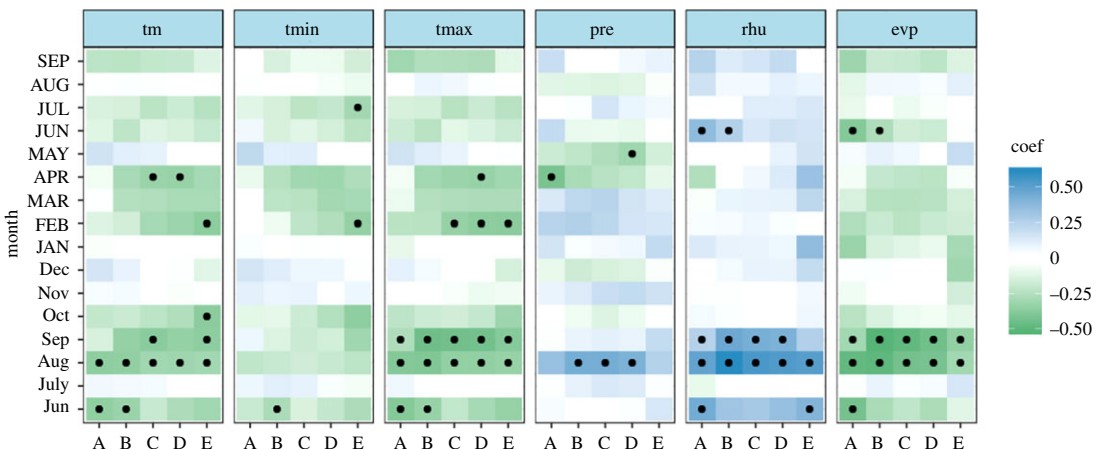

**Figure 6.** Temporally stationary plot relating the TR of Masson pine to climate drivers from the previous June to the current September. The moving correlation analysis was carried out in windows of 35 years, offset by 4 years. Significant correlations (*p* < 0.05) are denoted by black dots. 'Jun' denotes the previous June, 'JUN' denotes the current June; the time periods are 'A' (AD 1965–1999), 'B' (AD 1968–2002), 'C' (AD 1971–2005), 'D' (AD 1974–2008) and 'E' (AD 1977–2011).

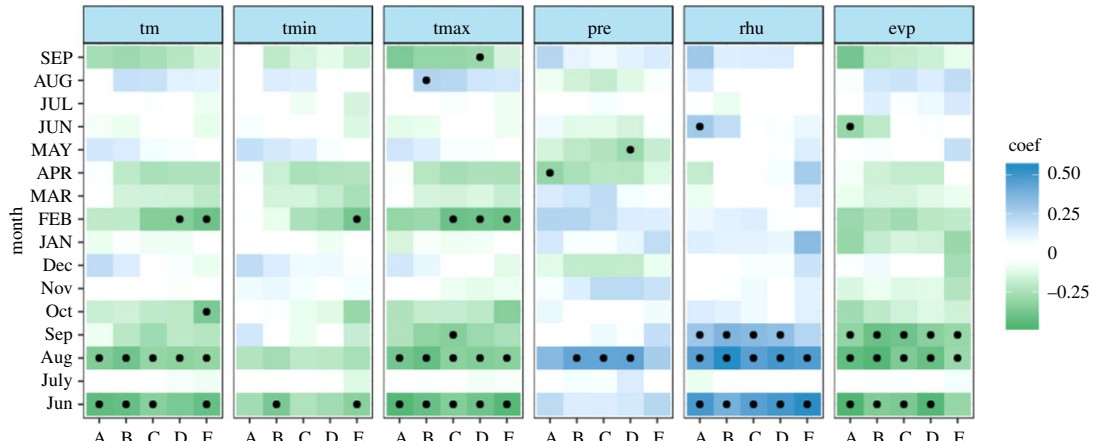

**Figure 7.** Temporally stationary plot relating the EW of Masson pine to climate drivers from the previous June to the current September. The moving correlation analysis was carried out in windows of 35 years, offset by 4 years. Significant correlations (*p* < 0.05) are denoted by black dots. 'Jun' denotes the previous June, 'JUN' denotes the current June; the time periods are 'A' (AD 1965–1999), 'B' (AD 1968–2002), 'C' (AD 1971–2005), 'D' (AD 1974–2008) and 'E' (AD 1977–2011).

patterns were found for these two TRW proxies. In summary, the temperature is the limiting factor, while moisture factors (precipitation, relative humidity) can promote the annual growth of Masson pine. Additionally, according to the absolute values of the correlation coefficients relative humidity, especially that in the previous August, exerts a significantly positive impact on tree-ring growth. On the other hand, climate drivers in the previous year seem to have a great influence on the TR and EW, whereas LW may be primarily controlled by the current temperature and moisture environment.

### 3.3.2. The temporal stability of dendroclimatic relationships between tree-ring width and monthly climate drivers

The majority of the correlation coefficients changed signs over time, and only a few monthly climate drivers revealed temporal stability at a significant level. TR and EW showed significant negative relationships with climate drivers in the previous August (mean temperature, maximum temperature and evaporation) and significant stable positive relationships with the relative humidity in the previous August (figures 6 and 7). However, a weak correlation with monthly evaporation for the three TRW indices was shown in figure 5. It can be concluded that there is no sign of a time-independent 'divergence effect' on tree radial growth [61]. In addition, compared with TR and EW,

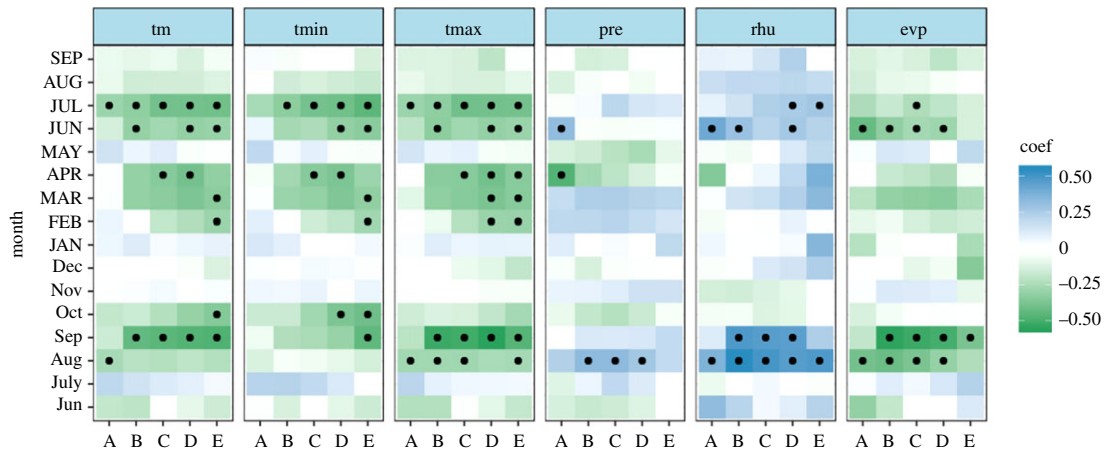

**Figure 8.** Temporally stationary plot relating the LW of Masson pine to climate drivers from the previous June to the current September. The moving correlation analysis was carried out in windows of 35 years, offset by 4 years. Significant correlations ($p < 0.05$) are denoted by black dots. 'Jun' denotes the previous June; 'JUN' denotes the current June; the time periods are 'A' (AD 1965–1999), 'B' (AD 1968–2002), 'C' (AD 1971–2005), 'D' (AD 1974–2008) and 'E' (AD 1977–2011).

LW exhibits obviously different behaviour in terms of the climate–response correlation and temporal stability. LW has a stable significant negative correlation with the mean and maximum temperature in the current July and the relative humidity in the previous August over the period AD 1961–2011 (figure 8). This is also in line with the correlation analysis for the monthly climate drivers (figure 5). For the monthly climate drivers that are significantly related to TRW parameters, although these relationships are not significant in some intervals of time, they do not change the sign of the correlation coefficient with tree growth over time. For example, the previous June mean temperature had a significant negative correlation with EW (figure 5), but the temporal stability test results indicated that the dendroclimatic correlation was not significant during the AD 1974–2008 period (figure 7). In summary, only the previous August relative humidity is significantly positively correlated with TR, EW and LW and shows a significant stable relationship with time. Considering the monthly climate–response patterns and temporal stability, TR and EW may be more sensitive to climate drivers in the previous year, whereas LW may be more susceptible to climate drivers in the current year.

## 3.4. The relationships between $\delta^{13}$C series and instrumental climate drivers

Only AC and HC were significantly and positively correlated with the current July mean temperature. The interannual variability of the stable carbon isotope ratio showed a generally positive correlation with the monthly minimum temperature, but lacked a significant response. The three components were strongly positively correlated with the current July maximum temperature. Clearly, it appears that the $\delta^{13}$C series of the three components (WW, AC and HC) have strong negative correlations with the July–August and December precipitation. The negative correlation coefficients for the July–September relative humidity are above the 0.05 level for WW, AC and HC. Additionally, these results (figure 9) showed a statistically significant relationship with summer growing season evaporation (July–September). According to Pearson's correlation analysis, the climate signal within the $\delta^{13}$C series is dominated by the climate in late summer and early autumn. These results indicate that the mean $\delta^{13}$C records for the three components are a better predictor of the variation in local Masson pine trees with different climate drivers than the three TRW parameters.

## 3.5. Comparison of climate-proxy relationships between tree-ring width and $\delta^{13}$C series

Based on the results of the previous analysis (§3.2–3.4), the common climate driver (relative humidity) has significant effects on the various tree-ring proxies. Therefore, we compared the correlation between the relative humidity and the TRW parameters/$\delta^{13}$C series during their common period (AD 1982–2011) (table 4). The correlation between the $\delta^{13}$C series of different components and relative humidity increased with the length of the considered period, and the maximum correlation coefficient

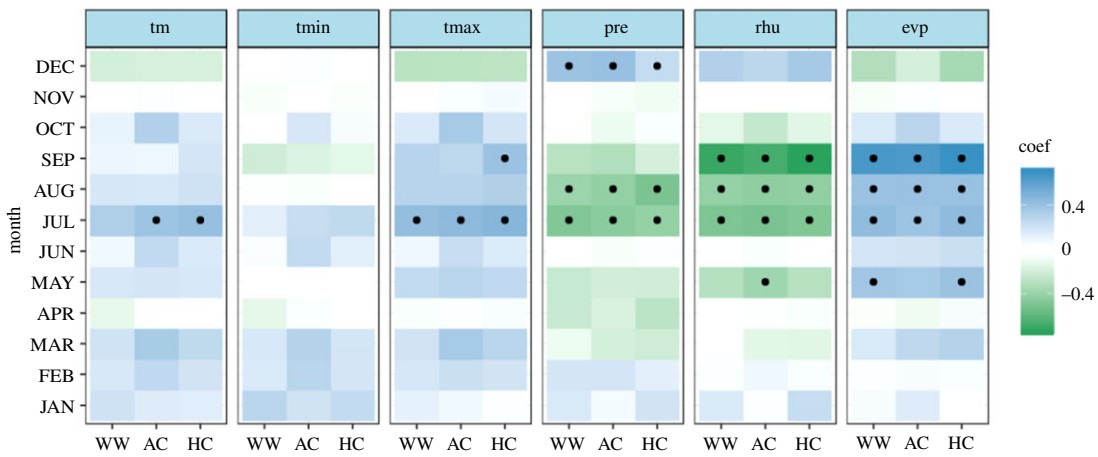

**Figure 9.** The correlation coefficients between the $\delta^{13}$C series (WW, AC and HC) and monthly climate drivers. Note: black dots indicate the significance level ($p < 0.05$). 'tm' denotes the mean temperature, 'tmin' denotes the minimum temperature, 'tmax' denotes the maximum temperature, 'pre' denotes the total precipitation, 'rhu' denotes the relative humidity and 'evp' denotes the evaporation. 'JUN' denotes the current June.

**Table 4.** The significant correlation between monthly relative humidity and tree-ring parameters (tree-ring width and $\delta^{13}$C series) during their common period (AD 1982–2011). (** denotes $p < 0.01$, * denotes $p < 0.05$, JUL–SEP indicates the period ranging from July to September in the current year, Jun–Aug indicates the period ranging from the previous June to the previous August).

|       | JUL       | AUG       | SEP       | JUL–SEP   | Jun      | Aug      | Jun–Aug  |
|-------|-----------|-----------|-----------|-----------|----------|----------|----------|
| TR    | —         | —         | —         | —         | 0.429**  | 0.522**  | 0.471**  |
| EW    | —         | —         | —         | —         | 0.530**  | 0.472**  | 0.514**  |
| LW    | —         | —         | —         | —         | 0.280    | 0.509**  | 0.375*   |
| WW    | −0.487**  | −0.461**  | −0.740**  | −0.769**  | —        | —        | —        |
| AC    | −0.506**  | −0.484**  | −0.694**  | −0.758**  | —        | —        | —        |
| HC    | −0.486**  | −0.473**  | −0.774**  | −0.792**  | —        | —        | —        |

occurred in three-month periods ending with the current September (WW (−0.769), AC (−0.758), HC (−0.792)). The seasonal correlation coefficients were all lower than −0.71. However, the correlation analysis results revealed that the climate signal became weak with the increasing length of the period considered for the TRW parameters. For example, TR had the strongest positive correlation coefficient (0.522) with the relative humidity in the previous August but revealed a slightly weakened climate signal (0.471) when considering the average previous June–August relative humidity. On the other hand, without considering the previous/current year, these results show that the $\delta^{13}$C series may be more sensitive to relative humidity than the TRW parameters during their common period (AD 1982–2011).

## 3.6. Spatial analysis

### 3.6.1. Spatial signature of tree-ring width parameters

The relative humidity value is proportional to the vapour pressure. Therefore, the vapour pressure field (http://climexp.knmi.nl) was spatially correlated with the TRW time series (figure 10). The proxies (TR, EW and LW) revealed similar hydroclimatic signal fingerprints for the previous June and August vapour pressure fields. The core area of the vapour pressure field shows quite clearly in our sampling sites. TR and EW revealed relatively high spatial correlation coefficients with the previous June vapour pressure ($r > {\sim}0.4$, $p < 0.01$), but LW showed a weaker correlation. In addition, when the TR and EW signals were correlated with the vapour pressure field of the previous June, the significantly correlated region mainly expanded to the southwest, whereas LW showed a relatively restricted zone. However, in the case of the

**Figure 10.** The spatial correlation between vapour pressure (previous June and August) and the three tree-ring width parameters. The first row of the graph indicates the spatial signature between the previous June vapour pressure and TR (a), EW (b) and LW (c). The second row of the graph indicates the spatial signature between the previous August vapour pressure and TR (d), EW (e) and LW (f).

vapour pressure in the previous August, the significantly positively correlated pole area ($r < \sim 0.4$, $p < 0.05$) is far from the sampling sites and covers a much smaller area.

### 3.6.2. Spatial signature of the $\delta^{13}$C series of three components

Because temperature revealed a weak relationship with the $\delta^{13}$C series of the different components, the spatial correlations between moisture drivers (vapour pressure and precipitation) and the three components (WW, AC and HC) were determined and are shown in figure 11. We found that the validity zone of the spatial correlation coefficients between the $\delta^{13}$C series and the July–September vapour pressure is very small, and HC had no significant spatial correlation. However, the fingerprints of the hydroclimatic signals in the $\delta^{13}$C series of the different components in the July–September precipitation field are generally similar. The core area of the vapour pressure field ($r < \sim -0.5$, $p < 0.01$) shows quite clearly in our sampling sites. The larger significant correlation regions and the similar spatial distribution patterns of the three components suggest that they represent the best precipitation proxy. In the southeastern region of the study area, the $\delta^{13}$C series of the three components are significantly and positively correlated with the July–September precipitation field.

## 4. Discussion

### 4.1. The potential of total ring width, earlywood width and latewood width for dendroclimatology

Generally, the mean sensitivity of tree rings ranges from 0.1 to 0.6 [40]. The statistical characteristics of TR, EW and LW chronologies of Masson pine in this study area show that the mean sensitivity values fall within the ranges in existing research results [62–64], and all exceeded 0.15 but were less than 0.30. This may be related to the long-term adaptation of Masson pine to the warm and humid climate in the subtropical region. The pairwise correlation between EW and LW was tested. EW has a strong relationship with LW in our study, which supports the findings from another study in the southwestern United States [65]. This conclusion is also consistent with those for other tree species

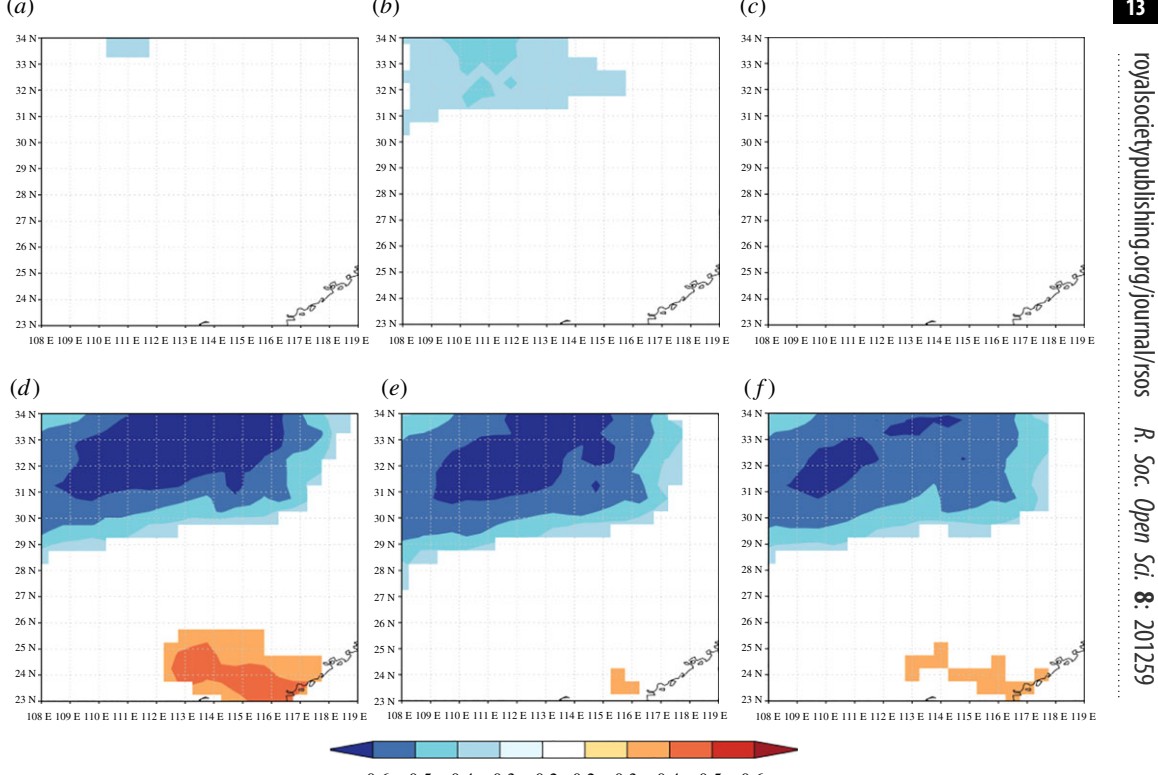

**Figure 11.** The spatial correlation between current July–September moisture drivers (vapour pressure and precipitation) and $\delta^{13}$C series of three components. The first row of this graph indicates the spatial signature between vapour pressure and WW ($a$), AC ($b$) and HC ($c$). The second row of this graph indicates the spatial signature between precipitation and WW ($d$), AC ($e$) and HC ($f$).

such as *Picea crassifolia* [66], *Larix gmelinii* [67] and *Cryptomeria fortunei* [68] in subtropical China. Consequently, according to other studies [69–71], the strong relationship between EW and LW indicates that the potential for reconstructing palaeoclimate by using the LW of Masson pine in subtropical China may be limited. Additionally, a comparison with the statistical characteristics of tree-ring chronologies shows that LW is relatively weakly sensitive to climate drivers (see §3.1). However, studies in Europe and the United States showed that LW was more sensitive to climate drivers than EW [22,25]. It is possible that the intensity of the correlation between EW and LW varies due to the temporal and spatial variability of the climate and the timber production time, the species specificity and age-related effects of growth on the cambium meristem. Some authors added another TRW index (i.e. adjusted tree-ring LW) to explore the climatic sensitivity [72]. The adjusted tree-ring LW is the residuals of the linear model between LW and EW. Since the impact of EW on LW has been removed, the adjusted LW can be regarded as the unique variation of LW. However, this TRW index did not reveal much stronger climate signals rather than TR, EW and LW in our work (electronic supplementary material, figure S2). Therefore, from this angle, it is not necessary to separate the EW and LW in Masson pine in subtropical China. Of course, Cao *et al*. [73] developed the first blue intensity series of EW and LW from Masson pine in Fujian province. This technique is worth trying in later research.

The Tongbai Mountains are located in a temperate zone with a continental monsoon climate. May–August is the wettest period, and it is relatively dry during the rest of the year. Across the whole distribution area of Masson pine, the annual precipitation is much lower than that in the southern and southeastern regions of subtropical China (figure 1*c*). Therefore, the TRW is positively correlated with the monthly precipitation from the previous June to the current March, but this correlation is not significant. Whereas, we found that monthly mean temperature and maximum temperature are negative factors for the growth of Masson pine, which is consistent with other research on climate reconstruction using this species in Macheng, Hubei Province [8]. The negative relationship between TRW and temperature may be caused by soil moisture status since high temperature could induce high evapotranspiration of soil moisture. Some research used the moisture status index like SPEI and scPDSI [74,75] because these indices are better to depict soil moisture status than precipitation, relative

humidity and evaporation. However, according to the time-varying bootstrapped correlation results, our research found that the relationships between LW and moisture status index are not stationary (electronic supplementary material, figure S3).

In the season of relatively low precipitation, from August to October, the relative humidity of the air is low. On the one hand, the longest daylight hours and the highest temperatures that occur during the June–August period accelerate soil water evapotranspiration. The higher air relative humidity inhibits transpiration in Masson pine, which leads to water stored in the leaves not being lost quickly. Additionally, climate drivers have an obvious influence on the 'lag' of tree growth in Masson pine. Consequently, the relative humidity of the previous year shows a positive relationship with the TRW chronologies. These results indicate that the higher moisture environment in the middle and late growing seasons promoted the radial growth of Masson pine. It is worth noting that the obviously upward temperature and slightly upward precipitation trends in the future may not be conducive to the radial growth of Masson pine.

## 4.2. Sensitivity comparison of different components of the $\delta^{13}$C series to climate parameters

Unlike recent studies, which reported that the values of $\delta^{13}$C were significantly correlated with temperature in subtropical China [76,77], temperature revealed a weak relationship with the $\delta^{13}$C series in our study. By contrast, despite abundant precipitation over the subtropical region, the main climate drivers that influenced the $\delta^{13}$C records of Masson pine in the subtropical region were precipitation and relative humidity. The most notable characteristic of the response of Masson pine to regional climate is the significant negative correlation between the precipitation and relative humidity in mid-summer and early autumn (July–September). Li *et al.* [78] also found that the $\delta^{13}$C sequence of Masson pine in Fujian Province in subtropical China was positively correlated with the September–October relative humidity. Note that our study area (32°24′ N, 113°16′ E) is far away from the study area (25°25′ N, 117°56′ E) of Li *et al.* [78]. Therefore, the relative humidity may be a common climate driver for the growth of Masson pine in subtropical China. In the future, an expanded network of sampling sites across all of subtropical China will be studied. Our results agree with those of a previous study from similar tree habitats in subtropical China [2,79]. The coefficients of the seasonal dendroclimatic relationship for the $\delta^{13}$C series exceeded the 0.7 level in our study, which has been proposed as a useful indicator for palaeoclimate reconstruction [80,81]. The specific mechanism may be as follows: after July in each year, the precipitation significantly decreases, while the temperature is still high. Consequently, the high temperature and low rainfall led to a significant reduction in the relative humidity of the air in our study area. On the one hand, the decrease in relative humidity accelerates soil water evaporation and reduces the available water supply for Masson pine. Additionally, it leads to an increase in the water vapour pressure difference inside and outside of the leaves. Under the combined action of the two factors, Masson pine must reduce the stomatal conductance of its leaves to reduce water evapotranspiration. However, a decrease in the stomatal conductance of leaves leads to an increase in the resistance of $CO_2$ entering the leaves, a decrease in the $CO_2$ concentration between mesophyll cells and a decrease in the $CO_2$ required for photosynthesis [82]. When the mesophyll cells lack $^{12}CO_2$, they have to use $^{13}CO_2$ which is relatively difficult to assimilate [30]. As a result, when the relative humidity of the air decreases, an increase in $\delta^{13}$C values is observed. The climate signal in $\delta^{13}$C tends to be dominated by stomatal conductance. Thus, the $\delta^{13}$C series variations are influenced by air humidity and those variables that control soil moisture status [78,80]. In addition, there is a significant positive correlation between the sunshine hours and the growth of Masson pine. This may be due to the lower relative humidity caused by the longer sunshine hours and the less rainy weather. On the other hand, long periods of sunshine can provide sufficient energy sources for plant photosynthesis. Sunlight promotes the absorption and utilization of $^{13}CO_2$, which leads to an increase in $\delta^{13}$C values.

## 5. Conclusion

This study was conducted to study the possible response of Masson pine to past hydroclimatic environmental changes as well as to future global climate changes for forest management purposes in subtropical China. The statistical characteristics of tree-ring chronologies using three TRW parameters are presented and indicate that EW shows a similar environmental signal to TR. Common climate signals from the previous September mean temperature, the August–September maximum temperature,

and the June relative humidity were found among the three TRW parameters. At both regional spatial and temporal scales, the three TRW parameters only have a significant positive correlation with the previous summer (June–August) moisture condition. The EW of Masson pine reveals strong signals of past climate and environmental changes at high temporal resolution in contrast with LW. Although these signals can be used as a supplement to the alternative indicators used in subtropical climate change research, these results indicate that separate analyses of different TRW proxies are unnecessary for future regional dendroclimatic research in Masson pine.

A correlation analysis revealed a relatively strong response to July–September moisture conditions (precipitation, relative humidity and evaporation) for the $\delta^{13}$C series of different wood components from the total ring. However, the $\delta^{13}$C series lack a significant relationship with temperature factors, except that the July maximum temperature relationships with these three components have high correlation coefficients. These results clearly suggest that the dominant leaf physiological forcing factor for the discrimination of $^{13}$C in Masson pine is stomatal conductance rather than the photosynthetic rate. Our study confirmed that $\delta^{13}$C values from the WW of Masson pine trees, i.e. without extracting HC or AC, are appropriate for reconstructing regional climate drivers at interannual and longer time scales.

A significant stable correlation was found for TRW and the total ring $\delta^{13}$C series with relative humidity. However, TRW parameters are significantly positively correlated with the relative humidity in the previous year, and $\delta^{13}$C series are significantly negatively correlated with this climate driver in the current year. In addition, the R-squared values of the strongest climate-proxy correlation coefficients are smaller than 50%, indicating that the TRW parameters are slightly less sensitive than the $\delta^{13}$C series of the total ring to climate drivers in our study area. However, for the same climate driver (e.g. relative humidity), seasonal correlation analysis reveals that the $\delta^{13}$C series of the three components describe the strong climate signal during their common period. These results suggest that the $\delta^{13}$C series may be a more useful indicator of the past climate in subtropical China than the TRW parameters. In our study, to avoid a shortage of samples and the 'juvenile effect', the $\delta^{13}$C series were only selected for a period of 30 years, because some of the tree cores had much narrower annual growth rings for a long time. Fortunately, there have been technical advancements in measuring stable isotopes. Andreu-Hayles et al. [83] detailed a high yield cellulose extraction system for small WW samples. Therefore, in the future, our study will investigate the long-term $\delta^{13}$C series by using these methods. We evaluated the climate sensitivity of TRW from the perspective of tree-ring sections including whole sections (total tree-ring) and subsections (earlywood, latewood). However, for the $\delta^{13}$C series, we conducted the analysis based on the chemical components of the total ring (WW, HC and AC) rather than by sections. The reason is that the wood samples were too small to separate the earlywood and latewood for chemical treatment.

Data accessibility. The data used in this paper are available at the Dryad Digital Repository: https://datadryad.org/stash/dataset/doi:10.5061/dryad.r4xgxd29d [84].

Authors' contributions. J.W. gave the conceptualization of this study; H.G. designed this manuscript; L.M. and C.L. collected the data with logistical assistance; L.M. carried out the laboratory work; H.G. and L.M. participated in data analysis; H.G. wrote the manuscript and all authors provided editorial advice. All authors gave the final approval publication.

Competing interests. The authors state they have no conflict of interest.

Funding. This study was supported by National Natural Science Foundation of China (grant no. 41271204); the Social Science Foundation of Anhui, China (grant no. SK2016A0544).

Acknowledgements. We are grateful for the helpful and constructive comments of our anonymous reviewers.

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
