## [Peer Review File · Royal Society Open Science]

Review History

RSOS-191626.R0 (Original submission)

Review form: Reviewer 1

Is the manuscript scientifically sound in its present form?

Yes

Are the interpretations and conclusions justified by the results?

Yes

Is the language acceptable?

No

Do you have any ethical concerns with this paper?

No

Have you any concerns about statistical analyses in this paper?

Yes

Recommendation?

Major revision is needed (please make suggestions in comments)

Comments to the Author(s)

Dear editor and the authors,

Thank you very much for inviting me to review this manuscript. In this study, the author explored the climatic sensitivity of multiple tree-ring proxies of Masson Pine from the northern limit of subtropical China. Generally, I appreciate this work because dendroclimatic research in subtropical China is far from sufficient and conclusive. Masson Pine is a widely distributed conifer species in subtropical China and commonly used by dendroclimatologists. I think the conclusion of this study has implications for dendroclimatic research in subtropical China. However, there are many problems in this current manuscript. I suggest publication after a major revision.

1. Major Problem

1.1 Although I'm not a native English speaker, I think there are too many English writing mistakes (i.e. grammar error, unprofessional language) in this manuscript, making it difficult to understand what the author wants to express. I strongly recommend the author to correct these errors with the help of native English speaker or professional English editing company.

1.2 The author compared the climate sensitivity of tree-ring width from the perspective of tree-ring sections including whole section (total tree-ring) and subsections (earlywood, latewood). However, for tree-ring $\delta^{13}C$, the author did not conduct the analysis from the perspective of sections but the chemical components of total ring (wholewood, holocellulose and α -cellulose). This has caused two problems.

The first problem is that the current study seems to be incomplete because the climate response of $\delta^{13}C$ in different chemical components of earlywood and latewood were not provided. If these results are provided, the author can get a final conclusion that which tree-ring parameter contains the strongest climatic signal. I think the reason is that wood samples are too small to separate earlywood and latewood for chemical treatment respectively. If so, please clarify this limit in the text.

The second problem is that the author should be very careful to make the third conclusion since the author only studied the $\delta^{13}C$ of different components of total ring. The author found that there is no significant difference among the climate sensitivity of different components of total ring. But this may be not real for earlywood and latewood. Therefore, I suggest the author focusing on the first two questions listed in the last paragraph of Page 3. The first question is the comparison of climate sensitivity of different tree-ring width. The second question is the comparison of climate sensitivity of $\delta^{13}C$ in different chemical components in total tree-ring. Besides, the author can make a comparison between the climate sensitivity of tree-ring width and $\delta^{13}C$, but only limited to total ring.

1.3 I suggested the author adding another tree-ring width index, that is "adjusted tree-ring latewood width". The adjusted tree-ring latewood width is defined by Meko and Baisan

(2001) (Meko D M and Baisan C H. Pilot study of latewood-width of conifers as an indicator of variability of summer rainfall in the North American monsoon region[J]. International Journal of Climatology, 2001, 21(6): 697-708.), which are residuals of linear model between latewood width and earlywood width. Since the impact of earlywood width on latewood width has been

removed, the adjusted latewood width can be regarded as the unique variation of latewood width. The adjusted latewood width may contain much stronger climatic signals of current year.

1.4 Tree-ring $\delta^{13}\text{C}$ is not only impacted by climate factors, but also the atmospheric $\delta^{13}\text{C}$. Because the atmospheric $\delta^{13}\text{C}$ exhibited significant downward trend since the beginning of industry revolution, the author need to remove the impact of atmospheric $\delta^{13}\text{C}$ on tree-ring $\delta^{13}\text{C}$.

1.5 Other hydroclimatic factors such as SPEI (Standard Precipitation Evapotranspiration Index) and scPDSI (self-calibrated Palmer Drought Severity Index) should be added in this manuscript because these indices are better to depict soil moisture status than precipitation, relative humidity and evaporation.

1.6 Correlation analysis should be performed with the first-order difference of timeseries to check whether the significant correlation is caused by the secular trend or the high frequency variations of timeseries. If there is no significant correlation between the first-order difference of timeseries, interpretation about the climatic signals of tree-ring timeseries should be careful. In other words, the tree-ring timeseries did not contain climate signals.

2 Detail Problem

2.1 Abstract

2.1.1 The abstract is too long and should be more concise. After describing the scientific question using several brief sentences, the main results and conclusions should be presented immediately. For example, the sentence "Results showed that the mean sensitivity of three ring width parameters is higher (0.190-0.266)" (Line 20-21 Page 2) is not necessary since it is not the main conclusion of this manuscript.

2.1.2 Line 28-30 Page 2. Please be more specific that which components these correlation coefficients corresponding to.

2.1.3 Line 34-35 Page 2. Please emphasis that it is total ring that there is no need to extract cellulose.

2.1.4 Line 36-38 Page 2. I don't think spatial correlation analysis is an appropriate method to compare the climate sensitivity of different tree-ring proxies. This is because a single site study cannot represent a regional condition.

2.1.5 Line 39-41 Page 2. Please emphasis that it is total ring.

2.1.6 Line 42 Page 2. "A more representative climate proxy". Which climate factor? Hydroclimate?

2.1.7 Line 43-44 Page 2. "Masson pine likely to be primarily influenced". The radial growth of Masson Pine? or The $\delta^{13}\text{C}$ of Masson pine.

3.1 Introduction

3.1.1 Line 15 Page 3. I suggest using "TW" instead of "TR".

3.1.2 Line 20-40 Page 3. This part is confusing. The author thinks that the limitation of total-ring width for climatic reconstruction in subtropical China is that the climatic response of total-ring

width is site-dependent. This means that the author wants to find some tree-ring proxies in which the climatic signals are consistent across space. However, as far as I know, using earlywood and latewood width may not be able to solve this problem. The reason that dendroclimatologists using earlywood and latewood width is that the earlywood and latewood may width provide much stronger and more temporally stable climatic signals than total tree-ring width. Therefore, I suggest the author rewriting the limitations of total ring-ring width and using other citations.

3.1.3 Line 44 Page 3. the sentence “combining multiple tree ring proxies were increasingly used to enhancing the climate signals” should be deleted since the author did not perform this analysis.

3.1.4 Line 45-48 Page 3. I suggest a more detail description in this part.

4.1 Material and methods

4.1.1 Line 36 Page 4. “Stem discs” or “tree-ring core”? Please confirm!

4.1.2 Line 3 Page 5.. Please change “ $^{13}\text{CPDB}$ ” to “ $\delta^{13}\text{C}$ (PDB)”

4.1.3 Line 19-Line 22 Page 5. What does this sentence mean? Please rewrite.

4.1.4 Line 26-Line 30 Page 5. It is not necessary to mention the climate characteristic of distribution area of Masson Pine since there is only one study site in this study.

4.1.5 Line 13 Page 6. Why the author select 35-yr window. 21-yr window is commonly used.

4.1.6 Line 13-15 Page 6. Please show the time span of tree-ring width series using a table.

5.1 Results

5.1.1 I think the part 3.1 and part 3.2 in the text should not be regarded as the main results since statistic character can not be used to depict the climate sensitivity of tree-ring time series. If the author really wants to retain these two parts, I suggest being briefer.

5.1.2 Line 6 Page 9. “are significant negative correlation with”. This is a writing error. I found there are lots of writing errors when describing the correlation/relationship between two timeseries. Please carefully check and correct them.

5.1.3 Line 17 Page 9. Please change “sole” to “single”.

5.1.4 Line 18-21 Page 9. What does this sentence “However, it can be basically found that there is a positive relationship between the three width indices and monthly precipitation” mean?

5.1.5 Line 27-31 Page 9. Why the author described the correlation between tree-ring width and evaporation in a second paragraph?

5.1.6 Line 32 Page 9. The author concluded that temperature is the limiting factor of tree-ring width. I think the negative relationship between tree-ring width and temperature may be caused by soil moisture status since high temperature could induce high evapotranspiration of soil moisture. The author did not use the moisture status index like SPEI and scPDSI but choose evaporation. However, evaporation is measured from the water surface which is different from the soil conditions. So, I suggest the author adding the results about SPEI and scPDSI and then make a final conclusion.

5.1.7 Line 36-38 Page 9. Please be more specific when illustrating the difference among climate response of multi-tree-ring width proxies. This is the key result of this study, and should be more detail.

5.1.8 Section 4.2. The author mainly interpreted why tree-ring $\delta^{13}\text{C}$ contain the climatic signals, but not illustrated the difference among the climatic sensitivities of different tree-ring components.

Figures

Figure 6 and 7: "Plot of Plot"? Please correct.

Figure 10 and 11: Please change the scope of the figure to make the sampling site as the center of the figure.

Review form: Reviewer 2

Is the manuscript scientifically sound in its present form?

No

Are the interpretations and conclusions justified by the results?

No

Is the language acceptable?

No

Do you have any ethical concerns with this paper?

No

Have you any concerns about statistical analyses in this paper?

No

Recommendation?

Reject

Comments to the Author(s)

This manuscript reports a study of masson pine tree rings from subtropical China, a region for which our understanding of tree growth and tree-ring formation is still very poor, and therefore may be of interest for the scientific community. However, a number of problems and weaknesses make the manuscript as it stands not publishable, and a revision of the text to make it acceptable would require such an effort that a major revision is not sufficient. Thus, I recommend the paper be rejected.

- 1) The use of the English language is poor and makes mainly the introduction and the discussion very difficult to be read and understood.
- 2) The text needs an accurate editing.
- 3) Some conceptual mistakes, as, for example, at line 60, tree rings seem to have the advantage of relatively well crossdating ... either are statistically significantly cross-dated or not! and why comparing with other proxies?

- 4) Tree rings in subtropical regions are usually difficult to be identified and dated. No mention here about such difficulties. No difficulties found, or just omitted? This problem should be described and discussed.
- 5) The aims are not clear, and frankly the whole study absolutely not innovative.
- 6) Four trees are maybe acceptable for a study of tree-ring stable isotopes, but are very few samples in order to build a ring-width chronology, not acceptable. The authors should establish a chronology with 20 trees, two cores from each, and then crossdate them with the four samples used for stable isotopes.
- 7) One good example of very confusing concepts and writing is given at the top of page 5: absolutely not clear what is meant.

Decision letter (RSOS-191626.R0)

23-Dec-2019

Dear Dr Gu:

Manuscript ID RSOS-191626 entitled "Comparison of dendroclimatic relationship using multiple tree-ring proxies (tree ring width and $\delta^{13}\text{C}$) from Masson pine" which you submitted to Royal Society Open Science, has been reviewed. The comments from reviewers are included at the bottom of this letter.

In view of the criticisms of the reviewers, the manuscript has been rejected in its current form. However, a new manuscript may be submitted which takes into consideration these comments.

Please note that resubmitting your manuscript does not guarantee eventual acceptance, and that your resubmission will be subject to peer review before a decision is made.

Your resubmitted manuscript should be submitted by 21-Jun-2020. If you are unable to submit by this date please contact the Editorial Office.

Kind regards,
Lianne Parkhouse
Editorial Coordinator
Royal Society Open Science
openscience@royalsociety.org

on behalf of Dr Emily Lindsey (Associate Editor) and Jon Blundy (Subject Editor)
openscience@royalsociety.org

Editorial Comments to Author:

As you have been requested to edit the written English, you must provide proof that you have done so: acceptable proof includes a certificate of language-editing from a language editing service or a signed letter from a native speaker of English. If you do not provide this proof, your manuscript may be returned to you.

For information about language editing services endorsed by the Royal Society, please follow the link below:

<https://royalsociety.org/journals/authors/language-polishing/>

Associate Editor Comments to Author (Dr Emily Lindsey):

Both reviewers indicated that this study has the potential to contribute to scientific understanding of dendrochronological and dendroclimatic research in a poorly-known region (subtropical China), and as such could merit publication in Royal Society Open Science. However, the English writing of the manuscript is not clear enough to adequately evaluate this article as currently written. I therefore must reject this manuscript at this time, but leave open the possibility of considering a subsequent version written in clearer English.

Both reviewers also raised quite serious concerns about some of the scientific methods, analyses, and conclusions in the article, and these should also be addressed prior to resubmission.

Reviewers' Comments to Author:

Reviewer: 1

Comments to the Author(s)

Dear editor and the authors,

Thank you very much for inviting me to review this manuscript. In this study, the author explored the climatic sensitivity of multiple tree-ring proxies of Masson Pine from the northern limit of subtropical China. Generally, I appreciate this work because dendroclimatic research in subtropical China is far from sufficient and conclusive. Masson Pine is a widely distributed conifer species in subtropical China and commonly used by dendroclimatologists. I think the conclusion of this study has implications for dendroclimatic research in subtropical China. However, there are many problems in this current manuscript. I suggest publication after a major revision.

1. Major Problem

1.1 Although I'm not a native English speaker, I think there are too many English writing mistakes (i.e. grammar error, unprofessional language) in this manuscript, making it difficult to understand what the author wants to express. I strongly recommend the author to correct these errors with the help of native English speaker or professional English editing company.

1.2 The author compared the climate sensitivity of tree-ring width from the perspective of tree-ring sections including whole section (total tree-ring) and subsections (earlywood, latewood). However, for tree-ring $\delta^{13}\text{C}$, the author did not conduct the analysis from the perspective of sections but the chemical components of total ring (wholewood, holocellulose and α -cellulose). This has caused two problems.

The first problem is that the current study seems to be incomplete because the climate response of

$\delta^{13}\text{C}$ in different chemical components of earlywood and latewood were not provided. If these results are provided, the author can get a final conclusion that which tree-ring parameter contains the strongest climatic signal. I think the reason is that wood samples are too small to separate earlywood and latewood for chemical treatment respectively. If so, please clarify this limit in the text.

The second problem is that the author should be very careful to make the third conclusion since the author only studied the $\delta^{13}\text{C}$ of different components of total ring. The author found that there is no significant difference among the climate sensitivity of different components of total ring. But this may be not real for earlywood and latewood. Therefore, I suggest the author focusing on the first two questions listed in the last paragraph of Page 3. The first question is the comparison of climate sensitivity of different tree-ring width. The second question is the comparison of climate sensitivity of $\delta^{13}\text{C}$ in different chemical components in total tree-ring. Besides, the author can make a comparison between the climate sensitivity of tree-ring width and $\delta^{13}\text{C}$, but only limited to total ring.

1.3 I suggested the author adding another tree-ring width index, that is “adjusted tree-ring latewood width”. The adjusted tree-ring latewood width is defined by Meko and Baisan (2001) (Meko D M and Baisan C H. Pilot study of latewood-width of conifers as an indicator of variability of summer rainfall in the North American monsoon region[J]. *International Journal of Climatology*, 2001, 21(6): 697-708.), which are residuals of linear model between latewood width and earlywood width. Since the impact of earlywood width on latewood width has been removed, the adjusted latewood width can be regarded as the unique variation of latewood width. The adjusted latewood width may contain much stronger climatic signals of current year.

1.4 Tree-ring $\delta^{13}\text{C}$ is not only impacted by climate factors, but also the atmospheric $\delta^{13}\text{C}$. Because the atmospheric $\delta^{13}\text{C}$ exhibited significant downward trend since the beginning of industry revolution, the author need to remove the impact of atmospheric $\delta^{13}\text{C}$ on tree-ring $\delta^{13}\text{C}$.

1.5 Other hydroclimatic factors such as SPEI (Standard Precipitation Evapotranspiration Index) and scPDSI (self-calibrated Palmer Drought Severity Index) should be added in this manuscript because these indices are better to depict soil moisture status than precipitation, relative humidity and evaporation.

1.6 Correlation analysis should be performed with the first-order difference of timeseries to check whether the significant correlation is caused by the secular trend or the high frequency variations of timeseries. If there is no significant correlation between the first-order difference of timeseries, interpretation about the climatic signals of tree-ring timeseries should be careful. In other words, the tree-ring timeseries did not contain climate signals.

2 Detail Problem

2.1 Abstract

2.1.1 The abstract is too long and should be more concise. After describing the scientific question using several brief sentences, the main results and conclusions should be presented immediately. For example, the sentence “Results showed that the mean sensitivity of three ring width parameters is higher (0.190-0.266)” (Line 20-21 Page 2) is not necessary since it is not the main conclusion of this manuscript.

2.1.2 Line 28-30 Page 2. Please be more specific that which components these correlation coefficients corresponding to.

2.1.3 Line 34-35 Page 2. Please emphasize that it is total ring that there is no need to extract cellulose.

2.1.4 Line 36-38 Page 2. I don't think spatial correlation analysis is an appropriate method to compare the climate sensitivity of different tree-ring proxies. This is because a single site study cannot represent a regional condition.

2.1.5 Line 39-41 Page 2. Please emphasize that it is total ring.

2.1.6 Line 42 Page 2. "A more representative climate proxy". Which climate factor? Hydroclimate?

2.1.7 Line 43-44 Page 2. "Masson pine likely to be primarily influenced". The radial growth of Masson Pine? or The $\delta^{13}\text{C}$ of Masson pine.

3.1 Introduction

3.1.1 Line 15 Page 3. I suggest using "TW" instead of "TR".

3.1.2 Line 20-40 Page 3. This part is confusing. The author thinks that the limitation of total-ring width for climatic reconstruction in subtropical China is that the climatic response of total-ring width is site-dependent. This means that the author wants to find some tree-ring proxies in which the climatic signals are consistent across space. However, as far as I know, using earlywood and latewood width may not be able to solve this problem. The reason that dendroclimatologists using earlywood and latewood width is that the earlywood and latewood may width provide much stronger and more temporally stable climatic signals than total tree-ring width. Therefore, I suggest the author rewriting the limitations of total ring-ring width and using other citations.

3.1.3 Line 44 Page 3. the sentence "combining multiple tree ring proxies were increasingly used to enhancing the climate signals" should be deleted since the author did not perform this analysis.

3.1.4 Line 45-48 Page 3. I suggest a more detail description in this part.

4.1 Material and methods

4.1.1 Line 36 Page 4. "Stem discs" or "tree-ring core"? Please confirm!

4.1.2 Line 3 Page 5.. Please change " ^{13}C CPDB" to " $\delta^{13}\text{C}$ (PDB)"

4.1.3 Line 19-Line 22 Page 5. What does this sentence mean? Please rewrite.

4.1.4 Line 26-Line 30 Page 5. It is not necessary to mention the climate characteristic of distribution area of Masson Pine since there is only one study site in this study.

4.1.5 Line 13 Page 6. Why the author select 35-yr window. 21-yr window is commonly used.

4.1.6 Line 13-15 Page 6. Please show the time span of tree-ring width series using a table.

5.1 Results

5.1.1 I think the part 3.1 and part 3.2 in the text should not be regarded as the main results since

statistic character can not be used to depict the climate sensitivity of tree-ring time series. If the author really wants to retain these two parts, I suggest being briefer.

5.1.2 Line 6 Page 9. “are significant negative correlation with”. This is a writing error. I found there are lots of writing errors when describing the correlation/relationship between two timeseries. Please carefully check and correct them.

5.1.3 Line 17 Page 9. Please change “sole” to “single”.

5.1.4 Line 18-21 Page 9. What does this sentence “However, it can be basically found that there is a positive relationship between the three width indices and monthly precipitation” mean?

5.1.5 Line 27-31 Page 9. Why the author described the correlation between tree-ring width and evaporation in a second paragraph?

5.1.6 Line 32 Page 9. The author concluded that temperature is the limiting factor of tree-ring width. I think the negative relationship between tree-ring width and temperature may be caused by soil moisture status since high temperature could induce high evapotranspiration of soil moisture. The author did not use the moisture status index like SPEI and scPDSI but choose evaporation. However, evaporation is measured from the water surface which is different from the soil conditions. So, I suggest the author adding the results about SPEI and scPDSI and then make a final conclusion.

5.1.7 Line 36-38 Page 9. Please be more specific when illustrating the difference among climate response of multi-tree-ring width proxies. This is the key result of this study, and should be more detail.

5.1.8 Section 4.2. The author mainly interpreted why tree-ring $\delta^{13}\text{C}$ contain the climatic signals, but not illustrated the difference among the climatic sensitivities of different tree-ring components.

Figures

Figure 6 and 7: “Plot of Plot”? Please correct.

Figure 10 and 11: Please change the scope of the figure to make the sampling site as the center of the figure.

Reviewer: 2

Comments to the Author(s)

This manuscript reports a study of masson pine tree rings from subtropical China, a region for which our understanding of tree growth and tree-ring formation is still very poor, and therefore may be of interest for the scientific community. However, a number of problems and weaknesses make the manuscript as it stands not publishable, and a revision of the text to make it acceptable would require such an effort that a major revision is not sufficient. Thus, I recommend the paper be rejected.

- 1) The use of the English language is poor and makes mainly the introduction and the discussion very difficult to be read and understood.
- 2) The text needs an accurate editing.

- 3) Some conceptual mistakes, as, for example, at line 60, tree rings seem to have the advantage of relatively well crossdating ... either are statistically significantly coss-dated or not! and why comparing with other proxies?
- 4) Tree rings in subtropical regions are usually difficult to be identified and dated. No mention here about such difficulties. No difficulties found, or just omitted? This problem should be described and discussed.
- 5) The aims are not clear, and frankly the whole study absolutely not innovative.
- 6) Four trees are maybe acceptable for a study of tree-ring stable isotopes, but are very few samples in order to build a ring-width chronology, not acceptable. The authors should establish a chronology with 20 trees, two cores from each, and then crrossdate them with the four samples used for stable isotopes.
- 7) One good example of very confusing concepts and writing is given at the top of page 5: absolutely not clear what is meant.

Author's Response to Decision Letter for (RSOS-191626.R0)

See Appendix A.

RSOS-201259.R0

Review form: Reviewer 1

Is the manuscript scientifically sound in its present form?

Yes

Are the interpretations and conclusions justified by the results?

Yes

Is the language acceptable?

No

Do you have any ethical concerns with this paper?

No

Have you any concerns about statistical analyses in this paper?

No

Recommendation?

Major revision is needed (please make suggestions in comments)

Comments to the Author(s)

Dear editors and authors, thank you for assigning me the manuscript for reviewing again. I think this resubmitted manuscript has made a lots of improvement, but still needs to be modified.

1. The No. 6 comment provided by reviewer 1 suggested the authour to make correlation analysis using 1st-order difference time series. This is a basic analysis step in tree-ring study which must be performed.

2. The No. 15 comment provided by reviewer 1. The author did had modified this part but the English expresion shoud be improvided again.

3. For the introduction, I suggest the author to be more concised.

Review form: Reviewer 3 (Barbara Sensuła)

Is the manuscript scientifically sound in its present form?

Yes

Are the interpretations and conclusions justified by the results?

No

Is the language acceptable?

Yes

Do you have any ethical concerns with this paper?

No

Have you any concerns about statistical analyses in this paper?

Yes

Recommendation?

Major revision is needed (please make suggestions in comments)

Comments to the Author(s)

I have not finish to check this paper- probably there is a mistake in methodology, could you send the information about standard deviation of your resutls for d13C?

Decision letter (RSOS-201259.R0)

Dear Dr Gu

On behalf of the Editors, we are pleased to inform you that your Manuscript RSOS-201259 "Comparison of dendroclimatic relationships using multiple tree-ring indicators (tree ring width and $\delta^{13}C$) from Masson pine" has been accepted for publication in Royal Society Open Science subject to minor revision in accordance with the referees' reports. Please find the referees' comments along with any feedback from the Editors below my signature.

Please ensure you seek additional advice from a service listed at <https://royalsociety.org/journals/authors/benefits/language-editing/> before resubmitting your revision.

Please submit your revised manuscript and required files (see below) no later than 7 days from today's (ie 12-Apr-2021) date. Note: the ScholarOne system will 'lock' if submission of the revision is attempted 7 or more days after the deadline. If you do not think you will be able to meet this deadline please contact the editorial office immediately.

on behalf of Dr Emily Lindsey (Associate Editor)
openscience@royalsociety.org

Associate Editor Comments to Author (Dr Emily Lindsey):

Comments to the Author:

Reviewers noted an improvement over the previous version of the manuscript, but still raised concerns over data analysis, presentation, and language; revisions should address these concerns.

Reviewer comments to Author:

Reviewer: 1

Comments to the Author(s)

Dear editors and authors, thank you for assigning me the manuscript for reviewing again. I think this resubmitted manuscript has made a lots of improvement, but still needs to be modified.

1. The No. 6 comment provided by reviewer 1 suggested the authour to make correlation analysis using 1st-order difference time series. This is a basic analysis step in tree-ring study which must be performed.
2. The No. 15 comment provided by reviewer 1. The author did had modified this part but the English expreesion should be improvided again.
3. For the introduction, I suggest the author to be more concised.

Reviewer: 3

Comments to the Author(s)

I have not finish to check this paper- probably there is a mistake in methodology, could you send the information about standard deviation of your resutls for d13C?

===PREPARING YOUR MANUSCRIPT===

===PREPARING YOUR REVISION IN SCHOLARONE===

Author's Response to Decision Letter for (RSOS-201259.R0)

See Appendix B.

Decision letter (RSOS-201259.R1)

Dear Dr Gu

On behalf of the Editors, we are pleased to inform you that your Manuscript RSOS-201259.R1 "Comparison of dendroclimatic relationships using multiple tree-ring indicators (tree ring width and $\delta^{13}\text{C}$) from Masson pine" has been accepted for publication in Royal Society Open Science subject to minor revision in accordance with the referees' reports. Please find the referees' comments along with any feedback from the Editors below my signature.

Please submit your revised manuscript and required files (see below) no later than 7 days from today's (ie 18-May-2021) date. Note: the ScholarOne system will 'lock' if submission of the revision is attempted 7 or more days after the deadline. If you do not think you will be able to meet this deadline please contact the editorial office immediately.

on behalf of Dr Emily Lindsey (Associate Editor)
openscience@royalsociety.org

Associate Editor Comments to Author (Dr Emily Lindsey):
Associate Editor
Comments to the Author:

This is a solid paper representing novel contributions to scientific understanding of dendroclimatology and Masson pine ecology in subtropical China.

It still requires some minor language editing for clarity; the authors should review my comments throughout in the attached .pdf and make the requisite corrections before publishing.

===PREPARING YOUR MANUSCRIPT===

===PREPARING YOUR REVISION IN SCHOLARONE===

-- Ensure that your data access statement meets the requirements at <https://royalsociety.org/journals/authors/author-guidelines/#data>. You should ensure that you cite the dataset in your reference list. If you have deposited data etc in the Dryad repository, please only include the 'For publication' link at this stage. You should remove the 'For review' link.

Author's Response to Decision Letter for (RSOS-201259.R1)

See Appendix C.

Decision letter (RSOS-201259.R2)

Dear Dr Gu,

I am pleased to inform you that your manuscript entitled "Comparison of dendroclimatic relationships using multiple tree-ring indicators (tree ring width and $\delta^{13}C$) from Masson pine" is now accepted for publication in Royal Society Open Science.

You can expect to receive a proof of your article in the near future. Please contact the editorial office (openscience@royalsociety.org) and the production office (openscience_proofs@royalsociety.org) to let us know if you are likely to be away from e-mail

contact – if you are going to be away, please nominate a co-author (if available) to manage the proofing process, and ensure they are copied into your email to the journal. Due to rapid publication and an extremely tight schedule, if comments are not received, your paper may experience a delay in publication.

on behalf of Dr Emily Lindsey (Associate Editor)
openscience@royalsociety.org

Appendix A

Dear editor and my reviewer,

Thank you for taking the time to review my paper. As a result of the significant disruption that is being caused by the COVID-19 pandemic, I have difficulty in meeting the timelines associated with resubmitting my paper. I really appreciate all your comments and suggestions! Your opinion has made great progress for me. All of your questions were answered below.

Sincerely,

Hongliang Gu

2020.07.15

Responses to reviewer 1

Comment 1:

1.1 Although I'm not a native English speaker, I think there are too many English writing mistakes (i.e. grammar error, unprofessional language) in this manuscript, making it difficult to understand what the author wants to express. I strongly recommend the author to correct these errors with the help of native English speaker or professional English editing company.

Reply: And sorry for the language problem, we had improved the language using a language editing service. The language editing certificate has been submitted in the system.

Comment 2:

1.2 The author compared the climate sensitivity of tree-ring width from the perspective of tree-ring sections including whole section (total tree-ring) and subsections (earlywood, latewood). However, for tree-ring $\delta^{13}C$, the author did not conduct the analysis from the perspective of sections but the chemical components of total ring (wholewood, holocellulose and α -cellulose). This has caused two problems.

The first problem is that the current study seems to be incomplete because the climate response of $\delta^{13}C$ in different chemical components of earlywood and latewood were not provided. If these results are provided, the author can get a final conclusion that which tree-ring parameter contains the strongest climatic signal. I think the reason is that wood samples are too small to separate earlywood and latewood for chemical treatment respectively. If so, please clarify this limit in the text.

The second problem is that the author should be very careful to make the third conclusion since the author only studied the $\delta^{13}C$ of different components of total ring. The author found that there is no significant difference among the climate sensitivity of different components of total ring. But this may be not real for earlywood and latewood. Therefore, I suggest the author focusing on the first two questions listed in the last paragraph of Page 3. The first question is the comparison of climate sensitivity of different tree-ring width. The second question is the comparison of climate sensitivity of $\delta^{13}C$ in different chemical components in total tree-ring. Besides, the author can make a comparison between the climate sensitivity of tree-ring width and $\delta^{13}C$, but only limited to total ring.

Reply: Dear, reviewer, we are grateful for your very good suggestion. For the first problem, in our study, wood samples are too small to separate earlywood and latewood for chemical

treatment respectively. Therefore, the climate response of $\delta^{13}\text{C}$ in different chemical components of earlywood and latewood were not provided. So, we have clarify this limit in the new manuscript at Line. Andreu-Hayles et al. (2019) developed a high-yield cellulose extraction system for small whole wood samples [1]. This method will be an appropriate choice for future stable isotope dendroclimatic proxy record studies. Based on this methods, in our feature work, the climate response of $\delta^{13}\text{C}$ in different chemical components of earlywood and latewood from Masson pine will be studied.

For the second problem, due to the insufficient sample for the $\delta^{13}\text{C}$ determination from earlywood and latewood, in my new manuscript, we deducted the comparison of climate sensitivity of different tree-ring width and different carbon isotope components from total ring, and made a comparison between the climate sensitivity of tree-ring width and $\delta^{13}\text{C}$ from total ring at the common time span.

Comment 3:

1.3 I suggested the author adding another tree-ring width index, that is “adjusted tree-ring latewood width”. The adjusted tree-ring latewood width is defined by Meko and Baisan (2001) (Meko D M and Baisan C H. Pilot study of latewood-width of conifers as an indicator of variability of summer rainfall in the North American monsoon region[J]. International Journal of Climatology, 2001, 21(6): 697–708.), which are residuals of linear model between latewood width and earlywood width. Since the impact of earlywood width on latewood width has been removed, the adjusted latewood width can be regarded as the unique variation of latewood width. The adjusted latewood width may contain much stronger climatic signals of current year.

Reply: Dear reviewer, thank you for your very good advice. In the previous research work, We have using the ‘adjusted tree-ring latewood width’ by using the residuals of linear model between latewood width and earlywood width in R language (i.e. lm function in base package). However, the adjusted latewood width contain less climatic signals (Fig.S1). From the figure1, the adjusted latewood width was significant correlation with mean temperature (i.e. September of previous, March, June, July of current year). Overall, temperature maybe a limiting climate factor for this tree ring parameter of Masson pine.

Figure S1. The bootstrapped correlation coefficients between TRW and monthly climate drivers of CRUs 4.02 datasets ($0.5^\circ \times 0.5^\circ$) (“TR” denotes the total ring width, “EW” denotes the earlywood width, “LW” denotes the latewood width, “ADJ” denotes the adjusted latewood width)

Comment 4:

1.4 Tree-ring $\delta^{13}C$ is not only impacted by climate factors, but also the atmospheric $\delta^{13}C$. Because the atmospheric $\delta^{13}C$ exhibited significant downward trend since the beginning of industry revolution, the author need to remove the impact of atmospheric $\delta^{13}C$ on tree-ring $\delta^{13}C$.

Reply: we have removed the impact of atmospheric $\delta^{13}C$ on tree-ring $\delta^{13}C$ by the mathematical method[2,3]. After mathematical correction caused by atmospheric $\delta^{13}C$ depletion, the r values between the raw and corrected $\delta^{13}C$ series ranged from 0.835 to 0.911. We found that the strength of climate sensitivity from raw and corrected $\delta^{13}C$ time series is most similarity.

Table S1. The correlation coefficient between climate factors and raw and corrected $\delta^{13}C$ time series

		Temperature		Precipitation		Relative humidity	
		raw	corrected	raw	corrected	raw	corrected
AC	8–9	0.272	0.406*	-0.578**	-0.531**	-0.672**	-0.651**
TBS	HC	0.255	0.413*	-0.437*	-0.429*	-0.630**	-0.701**
	WW	0.229	0.244	-0.528**	-0.469**	-0.663**	-0.667**

Comment 5:

1.5 Other hydroclimatic factors such as SPEI (Standard Precipitation Evapotranspiration Index) and scPDSI (self-calibrated Palmer Drought Severity Index) should be added in this manuscript because these indices are better to depict soil moisture status than precipitation, relative humidity and evaporation.

Reply: Dear reviewer, thank you for your good suggestion. In my previous work, the climate datasets from Xinyang meteorological station has high spatial resolution (i.e. $0.25^\circ \times 0.25^\circ$). In my opinion, this data set may be more accurate for dendroclimatology analysis. However, this dataset does not provide the SPEI and scPDSI data, which are better to depict soil moisture status than precipitation, relative humidity and evaporation. These two drought indices are sensitive to the available water content (AWC) of a soil type. Thus, applying these index for a climate division may be too general. The two soil layers within the water balance computations are simplified and may not be accurately representative of a location.

Of course, we can use the methods[4] to calculate these two indices. In the next work, we will add these two indices to conduct the dendroclimatology research on Masson pine.

Comment 6:

1.6 Correlation analysis should be performed with the first-order difference of timeseries to check whether the significant correlation is caused by the secular trend or the high frequency variations of timeseries. If there is no significant correlation between the first-order difference of timeseries, interpretation about the climatic signals of tree-ring timeseries should be careful. In other words, the tree-ring timeseries did not contain climate signals.

Reply: In fact, calculating the first order differencing of a time series is useful for converting a non stationary time series to a stationary form. Most climate time series are far from stationary when expressed in their original units of measurement, and even after deflation or seasonal adjustment they will typically still exhibit trends, cycles, random-walking, and other non-stationary behavior. If the series has a stable long-run trend and tends to revert to the trend line following a disturbance, it may be possible to stationarize it by de-trending (e.g., by fitting a trend line and subtracting it out prior to

fitting a model, or else by including the time index as an independent variable in a regression or ARIMA model), perhaps in conjunction with logging or deflating. Such a series is said to be trend-stationary.

Comment 7:

2.1.1 The abstract is too long and should be more concise. After describing the scientific question using several brief sentences, the main results and conclusions should be presented immediately. For example, the sentence “Results showed that the mean sensitivity of three ring width parameters is higher (0.190-0.266)” (Line 20-21 Page 2) is not necessary since it is not the main conclusion of this manuscript.

Reply: The abstract in my new manuscript has been modified. Some sentence has been eliminated.

Comment 8:

2.1.2 Line 28-30 Page 2. Please be more specific that which components these correlation coefficients corresponding to.

Reply: The phase ‘total ring’ has been appended to this sentence.

Comment 9:

2.1.3 Line 34-35 Page 2. Please emphasis that it is total ring that there is no need to extract cellulose.

Reply: We have modified this sentence.

Comment 10:

2.1.4 Line 36-38 Page 2. I don’t think spatial correlation analysis is an appropriate method to compare the climate sensitivity of different tree-ring proxies. This is because a single site study cannot represent a regional condition.

Reply: Dear reviewer, .

Comment 11:

2.1.5 Line 39-41 Page 2. Please emphasis that it is total ring.

Reply: We have emphasis this phase.

Comment 12:

2.1.6 Line 42 Page 2. “A more representative climate proxy”. Which climate factor? Hydroclimate?

Reply: This sentence has been rewriting in the new manuscript.

Comment 13:

2.1.7 Line 43-44 Page 2. “Masson pine likely to be primarily influenced”. The radial growth of Masson Pine? or The $\delta^{13}C$ of Masson pine.

Reply: This sentence has been rewritten.

Comment 14:

3.1.1 Line 15 Page 3. I suggest using “TW” instead of “TR”.

Reply: Dear reviewer, thank you for your suggestion, in some papers, ‘TW’ also was used.

Comment 15:

3.1.2 Line 20-40 Page 3. This part is confusing. The author thinks that the limitation of total-ring width for climatic reconstruction in subtropical China is that the climatic response of total-ring width is site-dependent. This means that the author wants to find some tree-ring proxies in which the climatic signals are consistent across space. However, as far as I know, using earlywood and latewood width may not be able to solve this problem. The reason that dendroclimatologists using earlywood and latewood width is that the earlywood and latewood may width provide much stronger and more temporally stable climatic signals than total tree-ring width. Therefore, I suggest the author rewriting the limitations of total ring-ring width and using other citations.

Reply: This part has been rewritten in my new manuscript: Compared with trees at middle and high latitudes or high elevations, the total ring width (TR) growing in mesic or humid areas has limitations for reflecting climate change at a large spatial scale. Therefore, the research of dendroclimatology in subtropical China are relatively under developed so far. Compared with TR, the earlywood (EW) and latewood (LW) formed in different tree ring growth season, which had higher temporal resolution than the TR (EW + LW), and might show more detailed characteristics of the tree ring climate relationship [5,6]. Some studies have shown that cross-dating was statistically more significant with LW compared with TR [7], and LW appeared to be strongest sensitivity to climatic drivers [5,6,8]. However, some researchers reported that the EW revealed strongest correlation with early summer hydroclimatic signal [9,10]

Comment 16:

3.1.3 Line 44 Page 3. the sentence “combining multiple tree ring proxies were increasingly used to enhancing the climate signals” should be deleted since the author did not perform this analysis.

Reply: We have deleted this sentence in our revised manuscript.

Comment 17:

3.1.4 Line 45-48 Page 3. I suggest a more detail description in this part.

Reply: In my new manuscript:

However, due to the warm and humid climate, subtropical forests and woodlands have a high diversity of developing plants, diverse wood anatomy, so the relationship between tree rings and climatic drivers is complex [11–13]. Compared with trees at middle and high latitudes or high elevations, the total ring width (TR) growing in mesic or humid areas has limitations for reflecting climate change at a large spatial scale. Therefore, the research of dendroclimatology in subtropical China are relatively under developed so far. Compared with TR, the earlywood (EW) and latewood (LW) formed in different tree ring growth season, which had higher temporal resolution than the TR (EW + LW), and might show more detailed characteristics of the tree ring climate relationship [5,6]. Some studies have shown that cross-dating was statistically more significant with LW compared with TR [7], and LW appeared to be strongest sensitivity to climatic drivers [5,6,8]. However, some researchers reported that the EW revealed strongest correlation with early summer hydroclimatic signal [9,10].

Comment 18:

4.1.1 Line 36 Page 4. “Stem discs” or “tree-ring core”? Please confirm!

Reply: We are grateful for the suggestion, and have corrected the “disc” into “corers” in the new manuscript.

Comment 19:

4.1.2 Line 3 Page 5.. Please change “13 CPDB” to “ $\delta^{13}C$ (PDB)”.

Reply: This phase has been changed.

Comment 20:

4.1.3 Line 19-Line 22 Page 5. What does this sentence mean? Please rewrite.

Reply: This sentence has been rewritten.

Comment 21:

4.1.4 Line 26-Line 30 Page 5. It is not necessary to mention the climate characteristic of distribution area of Masson Pine since there is only one study site in this study.

Reply: We are grateful for the suggestion, and we have deleted this parts of the climate characteristic of distribution area of Masson Pine.

Comment 22:

4.1.5 Line 13 Page 6. Why the author select 35-yr window. 21-yr window is commonly used.

Reply: In view of this problem, due to we use treeclim package for analysing the sensitivity of tree ring parameters. The default value of window size argument in dcc function is 25-yr. Some researchers used 51-yr window[14]; some authors used 31-yr window[15]. Overall, the window size To explore the time-dependency of proxy/climate relationships by applying response and correlation functions in moving window intervals, according to our attenuation needs, we select 35-yr window.

Comment 23:

4.1.6 Line 13-15 Page 6. Please show the time span of tree-ring width series using a table.

Reply: We have shown the time span of tree-ring width series in the new manuscript.

Comment 24:

5.1.1 I think the part 3.1 and part 3.2 in the text should not be regarded as the main results since statistic character can not be used to depict the climate sensitivity of tree-ring time series. If the author really wants to retain these two parts, I suggest being briefer.

Reply: We are grateful for the suggestion. This two part have been refined.

Comment 25:

5.1.2 Line 6 Page 9. “are significant negative correlation with”. This is a writing error. I found there are lots of writing errors when describing the correlation/relationship between two timeseries. Please carefully check and correct them.

Reply: We have change this writing error.

Comment 26:

5.1.3 Line 17 Page 9. Please change “sole” to “single”.

Reply: We have change this phase in the new paper.

Comment 27:

5.1.4 Line 18-21 Page 9. What does this sentence “However, it can be basically found that there is a positive relationship between the three width indices and monthly precipitation” mean?

Reply: Dear reviewer, I mean, tree ring width is positively correlated with precipitation in almost all months.

Comment 28:

5.1.5 Line 27-31 Page 9. Why the author described the correlation between tree-ring width and evaporation in a second paragraph?

Reply: At that time in the previous work, the evaporation in some months has significant

correlation with tree-ring width, so we described this results.

Comment 29:

5.1.6 Line 32 Page 9. The author concluded that temperature is the limiting factor of tree-ring width. I think the negative relationship between tree-ring width and temperature may be caused by soil moisture status since high temperature could induce high evapotranspiration of soil moisture. The author did not use the moisture status index like SPEI and scPDSI but choose evaporation. However, evaporation is measured from the water surface which is different from the soil conditions. So, I suggest the author adding the results about SPEI and scPDSI and then make a final conclusion.

Reply: We are grateful for the suggestion. However, from Fig S2, our research found that the sign of time-varying bootstrapped correlation coefficient of latewood width displays temporal fluctuations using these two indices (SPEI and scPDSI). because of the ‘divergence phenomenon’, i.e. the temporal instability of climate-tree growth relations since the end of the 20th century[16]. Consideration of the time-dependence that may affect dendroclimatic results is increasingly embedded into ecological studies. For comparison, the results about SPEI and scPDSI were not investigated in my new manuscript.

Figure S2. Temporally stationary plot relating the latewood width (LW) of Masson pine to climate drivers (CRU TS4.02 gridded dataset) from the previous June to the current September. The moving correlation analysis was carried out in windows of 35 yr, offset by 4 yr. Significant correlations ($p < 0.05$) are denoted by black dots. ‘Jun’ denotes the previous June, ‘JUN’ denotes the current June; the time periods are ‘A’ (1965–1999 AD), ‘B’ (1968–2002 AD), ‘C’ (1971–2005 AD), ‘D’ (1974–2008 AD), and ‘E’ (1977–2011 AD)

Figure S3. Temporally stationary plot relating the latewood width (EW) of Masson pine to climate drivers (CRU TS4.02 gridded dataset) from the previous June to the current September. The moving correlation analysis was carried out in windows of 35 yr, offset by 4 yr. Significant correlations ($p < 0.05$) are denoted by black dots. ‘Jun’ denotes the previous June, ‘JUN’ denotes the current June; the time periods are ‘A’ (1965–1999 AD), ‘B’ (1968–2002 AD), ‘C’ (1971–2005 AD), ‘D’ (1974–2008 AD), and ‘E’ (1977–2011 AD)

Figure S4. Temporally stationary plot relating the latewood width (TR) of Masson pine to climate drivers (CRU TS4.02 gridded dataset) from the previous June to the current September. The moving correlation analysis was carried out in windows of 35 yr, offset by 4 yr. Significant correlations ($p < 0.05$) are denoted by black dots. ‘Jun’ denotes the previous June, ‘JUN’ denotes the current June; the time periods are ‘A’ (1965–1999 AD), ‘B’ (1968–2002 AD), ‘C’ (1971–2005 AD), ‘D’ (1974–2008 AD), and ‘E’ (1977–2011 AD)

Comment 30:

5.1.7 Line 36-38 Page 9. Please be more specific when illustrating the difference among climate response of multi-tree-ring width proxies. This is the key result of this study, and should be more detail.

Reply: Dear reviewer, to investigate the potential of the multiple proxies (TR (EW+LW), EW, LW, WW, HC, and AC) of tree ring from Masson pine for the dendroclimatology studies in subtropical China. Questions to be answered are: (1) What climate drivers significantly affect the different tree ring width proxies of Masson pine? (2) whether differences exist in the

dendroclimatic relationships of the $\delta^{13}\text{C}$ series of different components? (3) For the total ring, make a comparison between the climate sensitivity of tree-ring width and $\delta^{13}\text{C}$.

Comment 31:

5.1.8 Section 4.2. The author mainly interpreted why tree-ring $\delta^{13}\text{C}$ contain the climatic signals, but not illustrated the difference among the climatic sensitivities of different tree-ring components.

Reply: The three components displayed uniform year-to-year variations and common significant climatic signals. The mean $\delta^{13}\text{C}$ record was significantly and negatively correlated with autumn hydroclimatic parameters (e.g., relative humidity and precipitation) in each study area.

Comment 32:

Figure 6 and 7: “Plot of Plot”? Please correct.

Reply: We have corrected this error.

Comment 33:

Figure 10 and 11: Please change the scope of the figure to make the sampling site as the center of the figure.

Reply: Dear reviewer, we explored the climatic sensitivity of multiple tree-ring proxies of Masson Pine from the northern limit of subtropical China (22° – 34°N , 98° – 122°E), so the spatial correlation were deducted at this scope of region. In our new revised manuscript, we do not change the scope of the figure to make the sampling site as the center of the figure.

Responses to reviewer 2

Comment 1:

The use of the English language is poor and makes mainly the introduction and the discussion very difficult to be read and understood.

Reply: I’m sorry to this language problem. We have using language edition service for improving our written English.

Comment 2:

The text needs an accurate editing.

Reply: We have done an accurate edition with the help of native English speaker.

Comment 3:

Some conceptual mistakes, as, for example, at line 60, tree rings seem to have the advantage of relatively well crossdating ... either are statistically significantly coss-dated or not! and why comparing with other proxies?

Reply: Due to the warm and humid climate, subtropical forests and woodlands have a high diversity of developing plants, diverse wood anatomy, so the relationship between tree rings and climatic drivers is complex [11–13]. Tree ring width are relatively insensitive to climate drivers (e.g. temperature) under a warm and humid climate in tropical and subtropical regions [17]. Which is better proxies for dendroclimatology in subtropical China? The dendroclimatic research in subtropical China is far from sufficient and conclusive.

Comment 4:

Tree rings in subtropical regions are usually difficult to be identified and dated. No mention here about such difficulties. No difficulties found, or just omitted? This problem should be described and discussed.

Reply: Tree rings in subtropical regions are usually difficult to be identified and dated. However, the annual ring boundaries of Masson pine are clear and easy to cross-date, and a number of results were achieved for climate reconstruction using Masson pine in the south-eastern part of China [18–23].

Comment 5:

Four trees are maybe acceptable for a study of tree-ring stable isotopes, but are very few samples in order to build a ring-width chronology, not acceptable. The authors should establish a chronology with 20 trees, two cores from each, and then crossdate them with the four samples used for stable isotopes.

Reply: We selected two sites from shady and sunny slopes for tree ring sampling. Overall, 65 trees (mean diameter at the breast height was ~1.3 m) were selected from edge trees or isolated trees with low canopy density in forests to reduce the impact of low-frequency changes in TRW series by tree competition. In addition, to carry out the splitting of the rings to obtain the stable carbon isotope signatures, tree ring cores (12 mm diameter increment) were obtained from four relatively old trees in the study area, that is relatively longer tree ring width chronologies [24]

Comment 6:

One good example of very confusing concepts and writing is given at the top of page 5: absolutely not clear what is meant.

Reply: We have corrected the confusing concepts in my whole paper.

Reference

1. Andreu-Hayles L *et al.* 2019 A high yield cellulose extraction system for small whole wood samples and dual measurement of carbon and oxygen stable isotopes. *Chemical Geology* **504**, 53–65. (doi:10.1016/j.chemgeo.2018.09.007)
2. McCarroll D *et al.* 2009 Correction of tree ring stable carbon isotope chronologies for changes in the carbon dioxide content of the atmosphere. *Geochimica et Cosmochimica Acta* **73**, 1539–1547. (doi:10.1016/j.gca.2008.11.041)
3. McCarroll D, Loader NJ. 2004 Stable isotopes in tree rings. *Quaternary Science Reviews* **23**, 771–801. (doi:10.1016/j.quascirev.2003.06.017)
4. Wells N, Goddard S, Hayes MJ. 2004 A Self-Calibrating Palmer Drought Severity Index. *Journal of Climate* **17**, 2335–2351. (doi:10.1175/1520-0442(2004)017<2335:ASPDSI>2.0.CO;2)
5. Koval I. 2013 Climatic Signal in Earlywood, Latewood and Total Ring Width of Crimean Pine (*Pinus nigra* subsp *pallasiana*) from Crimean Mountains, Ukraine. *Baltic For.* **19**, 245–251.

6. Lebourgeois F. 2000 Climatic signals in earlywood, latewood and total ring width of Corsican pine from western France. *Annals of Forest Science* **57**, 155–164. (doi:10.1051/forest:2000166)
7. Weigl M, Grabner M, Wimmer R. 2004 *Comparison of earlywood width, latewood width, and total ring-width measurements in oak*.
8. Yasue K, Funada R, Kobayashi O, Ohtani J. 2000 The effects of tracheid dimensions on variations in maximum density of *Picea glehnii* and relationships to climatic factors. *Trees* **14**, 223–229. (doi:10.1007/PL00009766)
9. Cabral-Alemán C, Pompa-García M, Acosta-Hernández A, Zúñiga-Vásquez J, Camarero J. 2017 Earlywood and Latewood Widths of *Picea chihuahuana* Show Contrasting Sensitivity to Seasonal Climate. *Forests* **8**, 173. (doi:10.3390/f8050173)
10. Zhao Y, Shi J, Shi S, Ma X, Zhang W, Wang B, Sun X, Lu H, Bräuning A. 2019 Early summer hydroclimatic signals are captured well by tree-ring earlywood width in the eastern Qinling Mountains, central China. *Clim. Past* **15**, 1113–1131. (doi:10.5194/cp-15-1113-2019)
11. Anhof D, Schleser GH. 2017 Tree ring studies in the tropics and subtropics. *Erdkunde* **71**, 1–4. (doi:10.3112/erdkunde.2017.01.06)
12. Bräuning A. 2011 Editorial note for the special issue on ‘Tropical Dendroecology’. *Trees* **25**, 1–2. (doi:10.1007/s00468-010-0530-x)
13. Brienen RJW, Schöngart J, Zuidema PA. 2016 Tree Rings in the Tropics: Insights into the Ecology and Climate Sensitivity of Tropical Trees. In *Tropical Tree Physiology* (eds G Goldstein, LS Santiago), pp. 439–461. Cham: Springer International Publishing. (doi:10.1007/978-3-319-27422-5_20)
14. Rydval M, Loader NJ, Gunnarson BE, Druckenbrod DL, Linderholm HW, Moreton SG, Wood CV, Wilson R. 2017 Reconstructing 800 years of summer temperatures in Scotland from tree rings. *Clim Dyn* **49**, 2951–2974. (doi:10.1007/s00382-016-3478-8)
15. Carrer M. 2011 Individualistic and Time-Varying Tree-Ring Growth to Climate Sensitivity. *PLoS ONE* **6**, e22813. (doi:10.1371/journal.pone.0022813)
16. D’Arrigo R, Wilson R, Liepert B, Cherubini P. 2008 On the ‘Divergence Problem’ in Northern Forests: A review of the tree-ring evidence and possible causes. *Global and Planetary Change* **60**, 289–305. (doi:10.1016/j.gloplacha.2007.03.004)
17. Shengtao L, Peng G, Panwei L, Xiang N, Bing W. 2019 The Response of Chinese Fir Forest Tree Ring Growth to Climate Change in China’s Dagangshan Region. *Pol. J. Environ. Stud.* **28**, 2371–2379. (doi:10.15244/pjoes/91788)
18. Cai Q, Liu Y. 2011 The June–September maximum mean temperature reconstruction from Masson pine (*Pinus massoniana* Lamb.) tree rings in Macheng, southeast China since 1879 AD.

Chinese Science Bulletin **55**, 2033–2039. (doi:10.1007/s11434-010-3235-z)

19. Kuang YW, Sun FF, Wen DZ, Zhou GY, Zhao P. 2008 Tree-ring growth patterns of Masson pine (*Pinus massoniana* L.) during the recent decades in the acidification Pearl River Delta of China. *Forest Ecology and Management* **255**, 3534–3540. (doi:10.1016/j.foreco.2008.02.036)
20. Chen F, Yuan Y, Wei W, Yu S, Zhang T. 2012 Reconstructed temperature for Yong'an, Fujian, Southeast China: Linkages to the Pacific Ocean climate variability. *Global and Planetary Change* **86–87**, 11–19. (doi:10.1016/j.gloplacha.2012.01.005)
21. Cai Q, Liu Y, Liu H, Sun C, Wang Y. 2017 Growing-season precipitation since 1872 in the coastal area of subtropical southeast China reconstructed from tree rings and its relationship with the East Asian summer monsoon system. *Ecological Indicators* **82**, 441–450. (doi:10.1016/j.ecolind.2017.07.012)
22. Luo D, Huang J-G, Jiang X, Ma Q, Liang H, Guo X, Zhang S. 2017 Effect of climate and competition on radial growth of *Pinus massoniana* and *Schima superba* in China's subtropical monsoon mixed forest. *Dendrochronologia* **46**, 24–34. (doi:10.1016/j.dendro.2017.08.001)
23. Li D *et al.* 2017 Climate, intrinsic water-use efficiency and tree growth over the past 150 years in humid subtropical China. *PLOS ONE* **12**, e0172045. (doi:10.1371/journal.pone.0172045)
24. Leavitt SW, Long A. 1984 Sampling strategy for stable carbon isotope analysis of tree rings in pine. *Nature* **311**, 145–147. (doi:10.1038/311145a0)

Appendix B

Dear editor and my reviewers,

Thank you for taking the time to review my paper. I really appreciate all your comments and suggestions! Your opinion has made great progress for me. All of your questions were answered below.

Sincerely,

Hongliang Gu

2021.04.19

Responses to editor

Comment 1:

Reviewers noted an improvement over the previous version of the manuscript, but still raised concerns over data analysis, presentation, and language; revisions should address these concerns.

Reply: Dear editor, the data analysis, presentation, and language have been addressed in my new manuscript. The language expression has been improved by a native speaker and a language-editing service.

Responses to reviewer 1

Comment 6:

1.6 Correlation analysis should be performed with the first-order difference of timeseries to check whether the significant correlation is caused by the secular trend or the high frequency variations of timeseries. If there is no significant correlation between the first-order difference of timeseries, interpretation about the climatic signals of tree-ring timeseries should be careful. In other words, the tree-ring timeseries did not contain climate signals.

Reply: We conducted the correlation analysis (Monte Carlo resamples 1000 times) with the first-order difference of timeseries (tree ring chronology and climate time series) . Differencing can help stabilise the mean of a non stationary time series by removing changes in the level of a time series, and therefore eliminating (or reducing) trend and seasonality. The result was showed at Fig S1. From the figure, we can see that the there is a significant correlation between the first-order difference of timeseries, and is not caused by the secular trend or the high frequency variations of timeseries. Considering the same datasets, we found that, different methods of analysis will produce different results. For example, the first-order difference of current June mean temperature timeseries is positively correlated with the first-order difference of tree ring timeseries (i.e. total ring width). Unfortunately, the correlation coefficient is negative between the raw timeseies. On the other hand, whether it's before or after the first-order difference of tree ring and previous summer mean temperature, the sign of correlation coefficient is the same. Therefore, we should pay attention to the inconsistency between the this two methods (i.e. first-order difference and no first-order difference of two timeseries).

In fact, calculating the first order differencing of a time series is useful for converting a non stationary time series to a stationary form. Most climate time series are far from stationary when expressed in their original units of measurement, and even after deflation or seasonal adjustment they will typically still exhibit trends, cycles, random-walking, and other non-stationary behavior. If the series has a stable long-run trend and tends to revert to the trend line following a disturbance, it may be possible to stationarize it by de-trending (e.g., by fitting a trend line and subtracting it out prior to

fitting a model, or else by including the time index as an independent variable in a regression or ARIMA model), perhaps in conjunction with logging or deflating. Such a series is said to be trend-stationary.

One way to determine more objectively whether differencing is required is to use a unit root test (Dickey-Fuller Test). These are statistical hypothesis tests of stationarity that are designed for determining whether differencing is required. We used the tseries package in R language to conduct the unit root test. The results of unit root test about some climate timeseries and tree ring timeseries reject the original hypothesis, that is to say these climate timeseries are stationary, and there is no need to do first-order difference. On the other hand, according to the google scholar, there are relatively few studies were rarely performed with the first-order difference of timeseries. We found one paper only conducted the first-order difference of tree ring chronologies[1]. Furthermore, to reduce the impact of low-frequency changes in the TRW series, trees (mean diameter at the breast height was ~1.3 m) were selected from edge trees or isolated trees with low canopy density in forests. Thanks my reviewer’s comments and suggestions. This is indeed a problem, and we will pay attention to this basic step in the later research.

In conclusion, in the subsequent analysis, we focus on the climate factors that whether before or after the first-order difference of timeseries, the sign of the correlation coefficient has not been changed. Additionally, one of the preliminary steps in reconstructions is the identification of the seasonal climate signal in the tree ring chronologies [2]. Moreover, compared with the correlation between tree ring proxies and monthly climate factors, seasonal or annual correlation analysis may explain the physiological significance of tree growth more clearly [3]. Therefore, it is necessary to integrate the results of seasonal correlation analysis to do dendroclimatology studies.

Fig S1. The correlation analysis results of first-order difference of tree ring and climate time series

Comment 15:

The author did had modified this part but the English expresion shoud be improvided again.

Reply: The parts in my new manuscript has been modified. The english expresion has been

improved by native speaker, please see the new manuscript or the following text:

Progress in dendrochronology techniques allow for using additional tree-ring indicators to enhance the climate signal and extend the reconstruction season or reconstruct different climate environmental variables [4]. For example, compared to TR, earlywood (EW) and latewood (LW) have higher temporal resolution than TR (EW + LW), and may show more valuable climate and environment signal [5,6]. Some studies have shown that cross-dating was statistically more significant with LW than with TR [7], and LW appeared to have the strongest sensitivity to climatic drivers [5,6,8]. Other researchers also reported that the EW revealed the strongest correlation with the early summer hydroclimatic signal [9,10].

Responses to reviewer 3

Comment 1:

I have not finish to check this paper- probably there is a mistake in methodology, could you send the information about standard deviation of your results for $\delta^{13}C$?

Reply: I'm sorry, but it's a pity. We supplement the information about the standard deviation of my results for $\delta^{13}C$ in the following table .

Table. 1 The statistic analysis of $\delta^{13}C$ records of different components within individual trees (“TBS01” denotes the sampling sites)

	TBS01			TBS02			TBS03			TBS04		
	WW	HC	AC	WW	HC	AC	WW	HC	AC	WW	HC	AC
mean/‰	-26.407	-24.977	-25.078	-25.699	-24.205	-24.297	-26.761	-25.560	-25.624	-25.970	-24.667	-24.615
median/‰	-26.426	-24.976	-25.010	-25.736	-24.301	-24.270	-26.864	-25.545	-25.786	-25.968	-24.555	-24.588
max/‰	-25.522	-23.692	-23.726	-24.593	-22.872	-23.144	-25.507	-24.160	-24.185	-25.083	-23.168	-23.299
min/‰	-26.920	-25.793	-26.294	-26.266	-24.948	-25.057	-27.490	-26.500	-26.750	-26.786	-25.839	-25.781
sd/‰	0.330	0.441	0.515	0.395	0.484	0.425	0.457	0.575	0.624	0.438	0.611	0.551

In addition, we also use the Welch'test and Bartlett.test approaches to assess the equality of variances. Results show that there is no significant difference for the same component between different trees at the same site. For the mean statistics of different components, the F-value and p-value of Welch'test results is 132.071, and less than 0.01 respectively. This results show that the mean statistics have significant difference among different components. Bartlett test results mean that there is no evidence to suggest that sd (standard deviation) in three components is different.

Reference

1. Brookhouse M, Brack C. 2006 Crossdating and analysis of eucalypt tree rings exhibiting terminal and reverse latewood. *Trees* **20**, 767–781. (doi:10.1007/s00468-006-0092-0)
2. Meko DM, Touchan R, Anchukaitis KJ. 2011 Seascorr: A MATLAB program for identifying the seasonal climate signal in an annual tree-ring time series. *Computers & Geosciences* **37**, 1234–1241. (doi:10.1016/j.cageo.2011.01.013)
3. Yang B, Qin C, Wang J, He M, Melvin TM, Osborn TJ, Briffa KR. 2014 A 3,500-year tree-ring record of annual precipitation on the northeastern Tibetan Plateau. *Proceedings of the*

National Academy of Sciences **111**, 2903–2908. (doi:10.1073/pnas.1319238111)

4. Arsalani M, Bräuning A, Pourtahmasi K, Azizi G, Mohammadi H. 2018 Multiple tree-ring parameters of *Quercus brantii* Lindel in SW Iran show a strong potential for intra-annual climate reconstruction. *Trees* **32**, 1531–1546. (doi:10.1007/s00468-018-1731-y)
5. Koval I. 2013 Climatic Signal in Earlywood, Latewood and Total Ring Width of Crimean Pine (*Pinus nigra* subsp *pallasiana*) from Crimean Mountains, Ukraine. *Baltic For.* **19**, 245–251.
6. Lebourgeois F. 2000 Climatic signals in earlywood, latewood and total ring width of Corsican pine from western France. *Annals of Forest Science* **57**, 155–164. (doi:10.1051/forest:2000166)
7. Weigl M, Grabner M, Wimmer R. 2004 *Comparison of earlywood width, latewood width, and total ring-width measurements in oak.*
8. Yasue K, Funada R, Kobayashi O, Ohtani J. 2000 The effects of tracheid dimensions on variations in maximum density of *Picea glehnii* and relationships to climatic factors. *Trees* **14**, 223–229. (doi:10.1007/PL00009766)
9. Cabral-Alemán C, Pompa-García M, Acosta-Hernández A, Zúñiga-Vásquez J, Camarero J. 2017 Earlywood and Latewood Widths of *Picea chihuahuana* Show Contrasting Sensitivity to Seasonal Climate. *Forests* **8**, 173. (doi:10.3390/f8050173)
10. Zhao Y, Shi J, Shi S, Ma X, Zhang W, Wang B, Sun X, Lu H, Bräuning A. 2019 Early summer hydroclimatic signals are captured well by tree-ring earlywood width in the eastern Qinling Mountains, central China. *Clim. Past* **15**, 1113–1131. (doi:10.5194/cp-15-1113-2019)

Appendix C

Dear editor,

Thank you for taking the time to review my paper. I really appreciate all your comments and suggestions! Your opinion has made great progress for me. I have review all your comments throughout in the attached .pdf and make the requisite corrections below.

Sincerely,

Hongliang Gu

2021.05.26

Responses to editor

Page 46 of 82, Line 20: delete the word 'is'.

Reply: I have delete this word in my new manuscript.

Page 46 of 82, Line 21: insert the word 'a'.

Reply: I have insert this word in my new manuscript.

Page 46 of 82, Line 21: convert the 'using' to 'use'.

Reply: I have done this correction in my new manuscript

Page 46 of 82, Line 31: convert the 'response' to 'responses'

Reply: I have insert this word in my new manuscript: Overall, the tree ring width parameters and the $\delta^{13}\text{C}$ series showed different responses to the same climate drivers during the same period.

Page 46 of 82, Line 32: delete the word 'the'.

Reply: I have delete this word in my new manuscript: the $\delta^{13}\text{C}$ series showed different responses to the same climate drivers during the same period.

Page 46 of 82, Line 51: convert the 'easy to copy' to 'easy replication'

Reply: I have done the correction in my new manuscript: According to the advantages of long-term records, high resolution, and widespread, tree rings have been proven to be a useful natural archive for climate change research.

Page 46 of 82, Line 52: delete the word 'the' in the sentence: "much progress has been made in studying climate change in the"

Reply: I have done the correction in my new manuscript: At present, dendroclimatology research works are increasingly carried out using tree-ring width in subtropical China.

Page 46 of 82, Line 59: delete the word 'the' in the sentence: "climate reconstructions were generated using Masson pine in the southeastern"

Reply: I have done the correction in my new manuscript: which is one of the most widely distributed and abundant tree species in subtropical China.

Page 47 of 82, Line 3: do you mean increase?

Reply: I have delete this sentence in my new manuscript

Page 47 of 82, Line 3: delete the “ was studied” in the sentence: Masson pine was studied in subtropical China.

Reply: I have delete this sentence in my new manuscript

Page 47 of 82, Line 12, convert the “formed” to “form” in the sentence: Unlike TR, EW and LW formed in different tree ring.

Reply: I have delete this sentence in my new manuscript.

Page 47 of 82, Line 14, convert the “season” to “seasons” in the sentence: growth season, and have higher temporal resolution than TR (EW + LW));

Reply: I have revised this sentence: compared to the total ring width (TR), earlywood width (EW) and latewood width (LW) have higher temporal resolution than TR (EW + LW) and may show more valuable climate and environment signals.

Page 47 of 82, Line 24: delete the “ the” in the sentence: for the paleoclimate, paleoenvironment

Reply: I have delete this word in my new manuscript.

Page 47 of 82, Line 29-34:This sentence needs to be re-written.

Reply: I have delete this word in my new manuscript: Studies have shown that $\delta^{13}\text{C}$ series of WW are better at extracting climate information than other organic compounds [32]. Some researchers believe that compared with other components, cellulose has a unique composition, and is easy to extract [35,36], and is more sensitive to climate change

Page 47 of 82, Line 39: delete the “ the” in the sentence: Thus, we investigated the potential of the multiple proxies

Reply: I have delete this word in my new manuscript.

Page 47 of 82, Line 40: delete the “ the” in the sentence: of Masson pine tree rings for the dendroclimatology studies in subtropical China.

Reply: I have delete this word in my new manuscript.

Page 48 of 82, Line 7, convert the “sunshine” to “daylight” in the sentence: The mild temperature and long sunshine time in summer are beneficial for the growth

Reply: I have done the correction in my new manuscript: The mild temperature and long daylight in summer are beneficial for

Page 48 of 82, Line 15, insert the word “and” in the sentence: At each sampling area (tongbaishan (32°24'N, 113°16'E), jigongshan (31°49'N, 114°03'E))

Reply: I have done the correction in my new manuscript: At each sampling area (Tongbaishan (32°24'N, 113°16'E), and Jigongshan (31°49'N

Page 48 of 82, Line 24, convert the “facilitatering” to “facilitate” in the sentence: were polished to facilitatering identification

Reply: I have done the correction in my new manuscript: cores were polished to facilitate ring identification

Page 48 of 82, Line 38, is this what you mean?

Reply: I have done the correction in my new manuscript: on clean glass plates for rasping. Four rasped increment cores of the same ring from per tree were cut, and mixed together.

Page 48 of 82, Line 42, convert the “samples” to “each sample” in the sentence: Part of the samples was collected for the

Reply: I have done the correction in my new manuscript: each sample was collected for the determination of the $\delta^{13}\text{C}$ signal of WW, and the rest was used

Page 48 of 82, Line 54, 57, 58: insert one space between two words

Reply: I have done the correction in my new manuscript: Therefore, the $\delta^{13}\text{C}$ values for each individual were measured, and the synthetic sequence of $\delta^{13}\text{C}$ in each study area was calculated using the arithmetic average method. Because the atmospheric $\delta^{13}\text{C}$ concentration has exhibited a significant downward trend since the beginning of the Industrial Revolution, we also mathematically removed the impact of atmospheric $\delta^{13}\text{C}$ on tree-ring $\delta^{13}\text{C}$

Page 49 of 82, Line 54, 55, delete the “the” and insert one space between two words in the sentence: difference of the timeseries to check whether the significant correlation was caused by the anactual trend or by the high frequency variations of the timeseries.

Reply: I have done the correction in my new manuscript: difference of the time series to check whether the significant correlation was caused by an actual

Page 50 of 82, Line 24, convert the “has” to “had” in the sentence: however LW has the lowest statistical values

Reply: I have done the correction in my new manuscript: generally high for each tree-ring parameter, whereas LW had the lowest statistical values

Page 50 of 82, Line 40, convert the “weaker” to “weaker” in the sentence: latewood width revealed a weaker relationship (0.88)

Reply: I have done the correction in my new manuscript: EW had the highest coefficient (0.94), and that between TR and LW revealed a weaker

Page 51 of 82, Line 28,30,33, 36,37, insert one space between two words in the sentence

Reply: I have done the correction in my new manuscript: All $\delta^{13}\text{C}$ sequences of WW were more depleted than those of HC and AC within individual trees according to two statistical indices (mean, standard deviation) (Table 3) because the lighter ligninin wood was extracted from the holocellulose samples but not from the WW samples. The values of WW ranged from -27.49‰ to -24.593‰, and AC and HC had ranges of -26.75 – -23.144‰ and -26.5 – -22.872‰, respectively. Although there were absolute differences in the values for WW, HC and AC, the sensitivity of different tree-ring components to climate change is evaluated based on the difference between the variation trends in the carbon isotope components and

the variation trends in the climatic drivers.

Page 53 of 82, Line 27, delete the “the” in the sentence: in the previous June can be seen compared with those of TR

Reply: I have done the correction in my new manuscript: and compared with TR (0.374, $p < 0.05$) and EW (0.481, $p < 0.05$), the relatively weak sensitivity of LW (0.173, $p < 0.1$) to the relative humidity in the previous June can be seen

Page 53 of 82, Line 36, explain the “positive factors” and rewrite a new sentence begin with word “according”

Reply: “moisture factors (precipitation, relative humidity) are the positive factors” means the moisture factors can promote the growth of Masson pine. I have done the correction in my new manuscript: while moisture factors (precipitation, relative humidity) can promote the annual growth of Masson pine. Additionally, according to the absolute values of the

Page 53 of 82, Line 59, convert the “indice” to “index” in the sentence: The bootstrapped correlation coefficients between TRW (TR, EW, LW, adjusted latewood width indice)

Reply: I have done the correction in my new manuscript: The bootstrapped correlation coefficients between TRW (TR, EW, LW) and monthly climate drivers from

Page 54 of 82, Line 14, delete the “]” in the sentence: negative relationships with climate drivers in the previous August] (mean temperature, maximum

Reply: I have delete the “]” in my new manuscript: negative relationships with climate drivers in the previous August (mean temperature, maximum

Page 54 of 82, Line 16, delete the “that” in the sentence: humidity in the previous August (Fig. 6, Fig. 7). However, Fig. 5 shows that a weak correlation

Reply: I have delete the “that” in my new manuscript: However, a weak correlation with monthly evaporation for the three tree-ring width indices was shown in Figure 5.

Page 57 of 82, Line 23, do you mean, “compared against”?

Reply: I have done the correction in my new manuscript: correlation. In addition, when the TR and EW signals were correlated with the vapor pressure field

Page 58 of 82, Line 46, insert the word “a” in the sentence: correlation between earlywood and latewood width was tested. Earlywood width has strong

Reply: I have done the correction in my new manuscript: LW was tested. EW has a strong relationship with LW in our study, which supports the findings

Page 59 of 82, Line 7, convert the word “explorer” to “explore”

Reply: I have done the correction in my new manuscript: latewood width) to explore the climatic sensitivity

Page 59 of 82, Line 8, of a linear model? of the linear model? of linear models?

Reply: I have **done the correction in my new manuscript: the residuals of the linear model between LW and EW.**

Page 59 of 82, Line 18, delete the word "the" in the sentence: The May-August is the wettest period, and it is relatively dry during the rest of the year.

Reply: I have **done the correction in my new manuscript: May-August is the wettest period, and it is relatively dry during the rest of the year.**

Page 59 of 82, Line 23, this sentence is unclear - rewrite.

Reply: I have **done the correction in my new manuscript: Whereas, we found that monthly mean temperature and maximum temperature are negative factors for the growth of Masson pine, which is consistent with other research on climate reconstruction using this species in Macheng, Hubei Province**

Page 59 of 82, Line 32, this sentence is confusing

Reply: I have **done the correction in my new manuscript: according to the time-varying bootstrapped correlation results, our research found that the the relationships between LW and moisture status index are not stationary**

Page 59 of 82, Line 36, convert the word "sunshine" to "daylight"

Reply: I have **done the correction in my new manuscript: On the one hand, the longest daylight hours and the highest temperatures that occur**

Page 59 of 82, Line 38, delete the word "the" in the sentence: humidity inhibits the transpiration in Masson pine,

Reply: I have **done the correction in my new manuscript: humidity inhibits transpiration in Masson pine, which leads to water stored in the leaves not being**

Page 59 of 82, Line 40, convert the word "driver" to "drivers" in the sentence: Additionally, climate driver have an obvious influence on the "lag" of tree

Reply: I have **done the correction in my new manuscript: Additionally, climate drivers have an obvious influence on the "lag" of tree growth in**

Page 60 of 82, Line 39, convert the word "expected" to "conducted" in the sentence: This study was expected to study the possible response of Masson pine to past hydroclimatic

Reply: I have **done the correction in my new manuscript: This study was conducted to study the possible response of Masson pine to past**

Page 60 of 82, Line 50, convert the "rather than" to "in contrast to" in the sentence: strong signals of past climate and environmental changes at high temporal resolution rather than

Reply: I have **done the correction in my new manuscript: in contrast to latewood width. Although these signals can be used as a supplement to the**

*Page 61 of 82, Line 4, convert the word “relaionships” to “relationships” in the sentence:
temperature factors, except that the July maximum temperature relaionships with these three*

**Reply: I have done the correction in my new manuscript: temperature factors, except that
the July maximum temperature relationships with these three**